# Raptor: Scalable Train-Free Embeddings for 3D Medical Volumes Leveraging Pretrained 2D Foundation Models

Ulzee An [*1]  Moonseong Jeong [*1]  Simon A. Lee [2]  Aditya Gorla [2 3]  Yuzhe Yang [1 2]  Sriram Sankararaman [1 2 4]

## Abstract

Current challenges in developing foundational models for volumetric imaging data, such as magnetic resonance imaging (MRI), stem from the computational complexity of training state-of-the-art architectures in high dimensions and curating sufficiently large datasets of volumes. To address these challenges, we introduce **Raptor** (Random Planar Tensor Reduction), a train-free method for generating semantically rich embeddings for volumetric data. Raptor leverages a frozen 2D foundation model, pretrained on natural images, to extract visual tokens from individual cross-sections of medical volumes. These tokens are then spatially compressed using random projections, significantly reducing computational complexity while retaining semantic information. Extensive experiments on ten diverse medical volume tasks verify the superior performance of Raptor over state-of-the-art methods, including those pretrained exclusively on medical volumes ($+3\%$ SuPreM, $+6\%$ MISFM, $+10\%$ Merlin, $+13\%$ VoCo, and $+14\%$ SLIViT), while entirely bypassing the need for costly training. Our results highlight the effectiveness and versatility of Raptor as a foundation for advancing deep learning-based methods for medical volumes (code: github.com/sriramlab/raptor).

## 1. Introduction

Understanding volumetric data is essential in a variety of applications, including healthcare (Milletari et al., 2016),

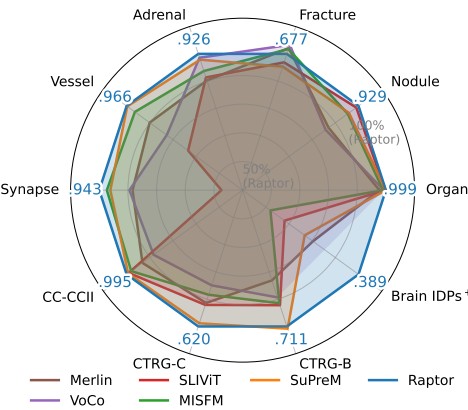

Figure 1. **Comparisons between Raptor and state-of-the-art methods on diverse medical volume tasks.** Raptor achieves superior performance across classification (AUROC↑) and regression (mean $r^2$↑, indicated with [+]) tasks while remaining entirely train-free.

human-computer interaction (Jiang et al., 2023), and autonomous systems (Huang et al., 2025). However, learning from volumetric modalities, represented as 3-dimensional voxels, meshes or point clouds, continues to pose logistical challenges compared with 2-D images. For instance, adapting convolutional (Yang et al., 2021) or transformer architectures designed for images (Dosovitskiy et al., 2021) to volumes remains computationally challenging as both classes of architectures incur cubic or higher order costs to learn or infer with volumes. Overcoming these hurdles generally requires the development of novel training objectives, efficient architectures, or access to large-scale compute infrastructure (though such infrastructure remains out of reach in most research settings). Furthermore, significant challenges remain in acquiring large-scale 3D training datasets relevant to the task of interest: the largest 3D medical datasets (160K volumes, Wu et al. 2024b) remain several orders of magnitude smaller than the 2D image datasets (1.2B images, Oquab et al. 2023) that have allowed researchers to scale large image models for current state-of-the-art 2D vision applications. As a result, few works to date have experimentally demonstrated the benefits of scaling large models for medical volumes (Zhu et al., 2023; Ma et al., 2024; Wu et al., 2024a), while large models for natural images and their capabilities are well-established (Radford et al., 2021;

---
[*]Equal contribution [1]Computer Science Department, University of California, Los Angeles [2]Department of Computational Medicine, David Geffen School of Medicine, University of California, Los Angeles [3]Bioinformatics Interdepartmental Program, University of California, Los Angeles [4]Department of Human Genetics, University of California, Los Angeles. Correspondence to: Ulzee An <ulzee@cs.ucla.edu>.

*Proceedings of the 42nd International Conference on Machine Learning*, Vancouver, Canada. PMLR 267, 2025. Copyright 2025 by the author(s).

Caron et al., 2021; Zhou et al., 2022; Saharia et al., 2022; Rombach et al., 2021; Ranftl et al., 2022; Kirillov et al., 2023; Liu et al., 2024a; Oquab et al., 2023).

In this work we propose **extracting semantically rich, low-dimensional embeddings from large 3D volumes by leveraging foundation models trained on generic 2D images** *without* **requiring training on the volumes**. The straightforward approach involving a direct application of an image foundation model (DINOv2-L, Oquab et al. 2023) to infer tokens over orthogonal cross-sections of a volume, however, leads to embeddings that are substantially larger than the volumes. To this end, we introduce **Raptor**: Random Planar Tensor Reduction, a *train-free* technique that drastically compresses the dimensionality of features over volumes by using *random projections*. Random projection is a class of scalable stochastic dimensionality reduction methods that map high-dimensional data into a lower-dimensional space by multiplying with a random matrix, with well-studied theoretical/empirical guarantees such as approximate preservation of pairwise distances and low-rank signals (Xie et al., 2017). We use random projections to compress tokens inferred by a frozen image foundation model over orthogonal cross-sections of a volume and then reduce major spatial dimensions to significantly compress the final representation. Our method reliably yields embeddings that have $\sim 99\%$ smaller footprint than that of raw voxels ($256^3$, compressed).

Despite its substantial size reduction, we show that Raptor embeddings retain volumetric information by demonstrating higher accuracies on downstream tasks than current state-of-the-art (SOTA) models, including those that leverage extensive pretraining (Figure 1). We demonstrate these capabilities without fine-tuning the underlying vision model or fitting any additional 3D architecture (i.e., *train-free*), making Raptor a particularly appealing choice in data-scarce and compute-constrained settings that are common in biomedical applications. Our work enables researchers with limited computational resources to utilize powerful pretrained models for high-dimensional volumetric analyses, thus broadening access to deep-learning-based 3D methods and fostering more inclusive innovation in machine learning.

Our contributions can be summarized as follows:

- Raptor is highly *data-efficient* – it is the first method that achieves state-of-the-art performance without requiring access to large datasets of medical volumes.
- Raptor embeddings are *train-free* – we eliminate the need for costly 3-dimensional model training, significantly reducing computational overhead.
- Raptor embeddings are *scalable* - they are *sub-cubic* with respect to input volume size, ensuring efficiency even for high-resolution medical volumes.

- Raptor is *model-agnostic* – it is designed to seamlessly integrate with and benefit from future advancements in image foundation models.

## 2. Related Works

Volumetric data span modalities such as point clouds (Caesar et al., 2020), meshes (Chang et al., 2015), and voxels (Yang et al., 2023). Each modality presents unique computational challenges; for instance, point clouds are not arranged in regular grids and have no per-point features such as color that are common for voxels (Sajid et al., 2025).

Raptor produces embeddings of voxels with cross sections, frequently obtained using magnetic resonance imaging (MRI) or computational tomography (CT). Below, we review works that learn useful representations from such medical volumes for downstream tasks.

**State-of-the-art architectures.** Many works have proposed adapting popular convolutional architectures, such as ResNet (He et al., 2016), for 3-dimensional data (Ning et al., 2019; Ebrahimi et al., 2020; Qayyum et al., 2021; Yang et al., 2021; Turnbull, 2022; Xue & Abhayaratne, 2023; Blankemeier et al., 2024). The success of the vision transformer (ViT) with shifted windows (Swin, Dosovitskiy et al. 2021; Liu et al. 2021) has inspired many recent works to build on ViT as the backbone network for 3D imaging tasks (Hatamizadeh et al., 2021; Wasserthal et al., 2023; Li et al., 2024; Cox et al., 2024; Wu et al., 2024b). However, transformer architectures incur a heavy computational cost in 3D due to self-attention[1]; subsequent works have proposed alternate architectures utilizing convolutional steps (Choy et al., 2019; Lai et al., 2024; Avram et al., 2024) or using forms of sub-quadratic attention (Liu et al., 2024b; Shaker et al., 2024; Xing et al., 2024; Dao, 2024).

**Large-scale pretraining for medical volumes.** Innovations in self-supervised learning (SSL) and the acquisition of web-scale datasets have enabled the development of foundational image (2D) models that demonstrate downstream capabilities that surpass domain-specific supervised models (Radford et al., 2021; Caron et al., 2021; Zhou et al., 2022; Saharia et al., 2022; Rombach et al., 2021; Ranftl et al., 2022; Kirillov et al., 2023; Liu et al., 2024a; Oquab et al., 2023). Several works have sought to replicate their success by collecting large datasets of medical volumes (3D) and scaling state-of-the-art methods.

Li et al. 2024 released a suite of models including Swin-UNETR, SegResNet, and ResNet trained on 5K CT volumes, collectively termed SuPreM, that are proposed to serve as foundation models given the pretraining process that involves 673K annotations over multiple organs. Simi-

---

[1]Liu et al. 2021 utilized $8 \times$V100 with batch size 1 per GPU

*Table 1.* **Current state-of-the-art models that leverage pretraining with medical volumes.** We state the scale of each method's medical volume pretraining data, its target medical domain, and its backbone architecture (*we note that Raptor is agnostic to any downstream domain, medical or not).

| METHODS | MEDICAL PRETRAINING DATA | DOMAIN | ARCHITECTURE |
|---|---|---|---|
| SLIViT (AVRAM ET AL., 2024) | 14M IMAGES (IMAGENET) +108K OCT IMAGES | OPTICAL | 2D CONV+3D TRANSFORMER |
| SuPreM (LI ET AL., 2024) | 5K CT VOLUMES | GENERAL | 3D SWINUNETR, 3D RESNET |
| MERLIN (BLANKEMEIER ET AL., 2024) | 15K VOLUMES | CHEST | 3D RESNET |
| MISFM (WANG ET AL., 2023) | 110K CT VOLUMES | GENERAL | 3D CONV+3D TRANSFORMER |
| VoCo (WU ET AL., 2024A) | 160K CT VOLUMES | GENERAL | 3D SWINUNETR |
| **RAPTOR** (OURS) | NONE (USES 2D FOUNDATION MODEL) | GENERAL* | 2D ViT + 3D COMPRESSION |

*Table 2.* **Sizes of comparable medical volume models.** We compare the parameter count of each medical volume model and the size of their latents. The latent was determined as the smallest bottleneck layer without pooling.

| METHODS | PARAM. COUNT | LATENT SIZE |
|---|---|---|
| SLIViT | 48.4M | $768 \times 64 \times 8 \times 8$ |
| SuPreM | 5.1M | $128 \times 12 \times 12 \times 12$ |
| MERLIN | 124.7M | $2048 \times 14 \times 7 \times 7$ |
| MISFM | 46.2M | $100 \times 16 \times 16 \times 16$ |
| VoCo | 294.9M | $3072 \times 3 \times 3 \times 3$ |
| **RAPTOR** (OURS) | 304.4M (DINOv2-L) | $3 \times 100 \times 16 \times 16$ ($K = 100$) |

larly, Wang et al. 2023 proposes MISFM, a medical image segmentation foundation model with a hybrid convolutional-transformer architecture trained on 110K scans containing multiple organs. Wu et al. 2024a proposes training a Swin-UNETR model with foundational capabilities called VoCo, based on a weakly-annotated dataset of 160K CT volumes that span multiple organs. This work takes advantage of a geometric SSL approach that leads to state-of-the-art capabilities on several downstream semantic segmentation tasks, outperforming SimCLR (Chen et al., 2020), Minkowski (Choy et al., 2019), UniMISS (Xie et al., 2024). Despite performance improvements, training these models still demands substantial computational resources.[2]

Other related works develop methods that are pretrained on specific CT modalities (e.g., chest) but still can be useful for general volumetric datasets when finetuned. Such approaches include Merlin (chest CT, Blankemeier et al. 2024) and SLIViT (optical CT, Avram et al. 2024). These methods leverage pretraining with up to $\sim 100$K volumes in their medical domains. We summarize notable prior works in Table 2.

**General-purpose embeddings for medical volumes.** A main feature of large-scale pretrained models is their ability to infer powerful latent representation of the target modality (i.e. embedding) that encompass features that generalize to many unseen downstream tasks (e.g. CLIP, Radford et al. 2021). In spite of ongoing efforts, few works qualify as being capable of generalizing to arbitrary medical volumes due to (1) the limited availability (in terms of size and access) of the 3D datasets and (2) the computational cost associated with scaling current approaches in 3D. To overcome these challenges, we propose a new paradigm to compute rich embeddings of volumes *without training any foundational models*.

## 3. Raptor

### 3.1. Overview of the approach

An outline of our approach is visualized in Figure 2. The core idea behind Raptor is to leverage a pretrained 2D image foundational model to encode the semantics of the 3D volumes in three different axial views. The resulting high-dimensional tensor is then aggregated over the axial views and low-rank approximated with random projections. We obtain the final Raptor embedding by flattening the three projections of the volumes. There is no training involved in any of the steps in our method.

### 3.2. 3D Inference using 2D foundation model

**Pretrained 2D foundation model.** We use the current state-of-the-art image foundation model as a domain agnostic feature extractor given 2-dimensional views of voxels. We utilize the latest DINOv2 model (Oquab et al., 2023) checkpoint at the time of writing[3]. DINOv2 consists of an image encoder, which is a vision transformer (ViT) trained on a curated dataset of 142M images (LVD-142M) and 1.2B additional images crawled on the web. The curated image dataset is reported to contain ImageNet-22k, the training split of ImageNet-1k, Google Landmarks and additional

---

[2]Wu et al. 2023 utilized $8\times$H100 GPUs for an undisclosed amount of time, while Li et al. 2024 discloses the use of $8\times$A100 GPUs for at least 7 days.

[3]https://huggingface.co/facebook/dinov2-large

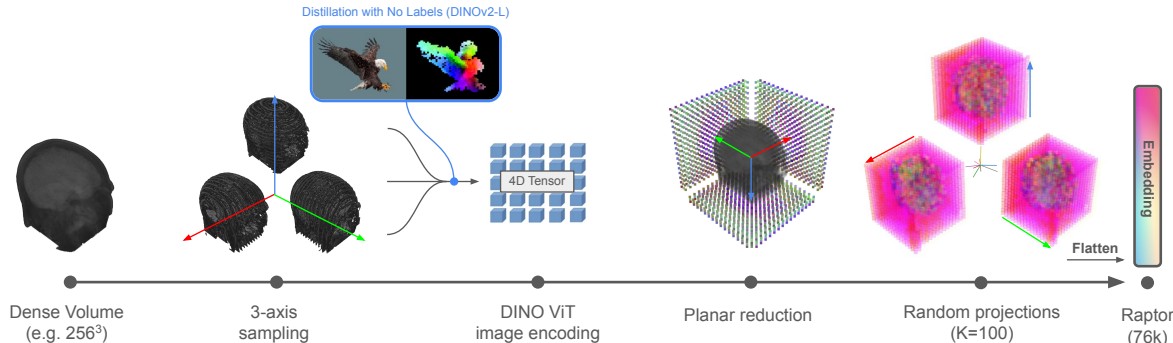

*Figure 2.* **Flowchart visualizing the computation of Raptor embeddings from a medical volume.** Raptor leverages a pretrained 2D image foundational model to encode the semantics of the 3D volumes in three different axial views. The resulting high-dimensional tensor is then low-rank approximated with random projections and aggregated. We obtain the final Raptor embedding by flattening the projections of the volumes. There is no training involved in any of the steps in our method (high-dimensional tensors not drawn to scale).

datasets containing natural images. The crawled images are described to have been identified via URL, making it unlikely for it to contain the training or test set of domain-specific datasets (such as Medical MNIST). We propose using the largest model (DINOv2-L) with 304M parameters (we validate this choice against other foundation models experimentally in A.1). We note that our approach is *agnostic* to the underlying choice of the image foundation model, and can benefit from continued development of better models. In A.3, we further demonstrate how features in the medical domain can be detected using a general purpose image encoder.

**3-axis volume sampling.** We apply DINOv2-L across 3D volumes along all three axes (axial, coronal, and sagittal – referring to top-down, front-back, and left-right directions; Figure A.1) of the volumes to capture cross-sectional features without relying on a 3-dimensional model. Given a volume of dimensions $256^3$, we apply the visual encoder of DINOv2-L to $3 \times 256$ total slices. The ViT defines a patch size $T$ (typically $T = 16$), resulting in an initial raw representation of $3 \times 256 \times 1024 \times 16 \times 16 \approx 201$M values, where $16 \times 16$ is the number of patches given slices of width and height $256 \times 256$ of the volume, and 1024 is the dimensionality of the patch embeddings.

Using the ViT to parse volumes in a dense fashion introduces computational challenges, as the inferred latent tensor is far greater than the size of the original image (in contrast to e.g., VAEs Kingma & Welling 2014). For instance, storing every tokenized slice of a volume requires 383MB of space using single precision floats (36TB+ for a dataset of 100K volumes), which is $127\times$ larger than the storage size of the $256^3$ cube in unsigned integers ($\sim 3$MB compressed with gzip, $\sim 300$GB for a dataset of 100k volumes).

### 3.3. Planar reduction using random projections

We describe the core contribution of Raptor, which is the efficient low-rank approximation of the high-dimensional tensor inferred by the DINOv2-L ViT through mean-pooling across the slice embeddings and random projections. This approach requires *no further training* of any parametric models, avoiding the need for additional GPU compute.

Given a volume $\boldsymbol{x} \in \mathbb{R}^{D \times D \times D}$, we specify all its slices (2-dimensional images) in three orthogonal directions as $\boldsymbol{S} \in \mathbb{R}^{3 \times D \times (D \times D)}$, where $\boldsymbol{S}_i, i \in \{1, 2, 3\}$ corresponds to $D$ slices of width and height $D \times D$ observed in one of the three directions (Figure A.1). We use a pretrained ViT $\phi(\cdot)$ that operates on images to obtain the intermediate representation $\boldsymbol{z} = \underset{1 \leq i \leq 3,}{\text{concat}}[\phi(\boldsymbol{S}_i)] \in \mathbb{R}^{3 \times D \times d \times p^2}$, where token size $d$ and number of patches $p = D/T$ are fixed by the ViT. We first reduce $\boldsymbol{z}$ by averaging the direction in which the slices were observed; repeating this for three views results in a tensor of size $3 \times d \times p^2$. Then, for each of the $p^2$ patches, we obtain a low-rank approximation of the spatially reduced tokens through random projections. For a number of projections $K$, we sample a projection matrix $\boldsymbol{R} \in \mathbb{R}^{K \times d}$, where $\boldsymbol{R}_{kl} \sim \mathcal{N}(0, 1)$ and $K < d$. Projecting the tokens with $\boldsymbol{R}$ results in a tensor of size $3 \times K \times p^2$. The final Raptor embedding is generated by flattening this tensor into a vector $\boldsymbol{v} \in \mathbb{R}^{3Kp^2}$:

$$\boldsymbol{v} = \text{flatten} \left( \underset{1 \leq i \leq 3}{\text{concat}} \left[ \boldsymbol{R} \frac{1}{D} \sum_{j=1}^{D} \boldsymbol{z}_{ij} \right] \right)$$

where $j$ indexes the tensor in $\boldsymbol{z}_{ij}$ corresponding to a slice in a particular axis $i$. A more detailed formulation of Raptor is described in Appendices A.5 and A.6. As described, the size of Raptor embeddings scale as $\mathcal{O}(p^2 K)$, which is sub-cubic in the dimension of the input volume, allowing us to store and use the representations for downstream tasks with ease.

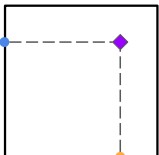 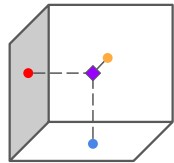

*Figure 3.* **Triangulation of a feature from a lower dimension.** Visualization of the intuition behind triaxial sampling. Features reduced to orthogonal lower dimensions can be used to triangulate features in the original space (e.g. in 2D or 3D).

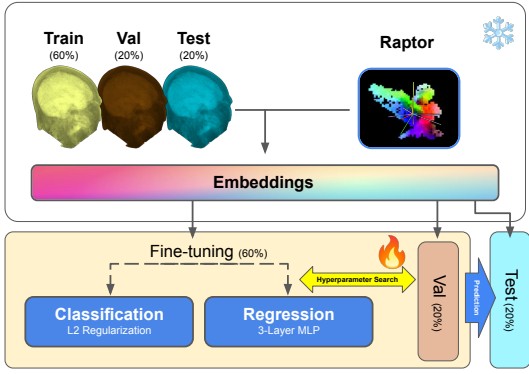

*Figure 4.* **Making downstream predictions using Raptor.** Raptor was used to obtain embeddings of medical volumes with no training by using a frozen image foundation model. For downstream tasks, embeddings of training and validation volumes were used to obtain the best lightweight predictor.

The quantities in typical settings are $D = 256, d = 1024, T = 16$, and $p = 256/16 = 16$ (using DINOv2-L), resulting in Raptor embeddings of size $768 \times K$ based on the choice of $K$. We infer two different Raptor embeddings on all datasets: **Raptor** using $K = 100$ and **Raptor-base** (**Raptor-B**) using $K = 10$ (determined experimentally, Sections 4.5 and 5.1), the latter of which is $10\times$ more compact.

**Effectiveness of random projections.** Random projection is underpinned by the Johnson–Lindenstrauss (JL) lemma (Dasgupta & Gupta, 2003), which states that distances between points (e.g., patch embeddings) are preserved up to a small distortion factor $(1 \pm \varepsilon)$ with high probability when mapped into $\mathbb{R}^K$ (see Appendix A.5 for a full proof). Empirically, we find that modest $K \approx 10$ preserves features that are comparable to state-of-the-art methods while reducing each slice embedding by up to $50\times$ or more (Section 5.1).

We choose random projection over other dimensionality reduction methods for two reasons. First, it allows for a more direct theoretical treatment (Appendix A.6). Second, given $N$ number of volumes, the time complexity of Raptor is $\mathcal{O}(p^2 dN(D + K))$ (A.7). In comparison, a full PCA to reduce to the same low-rank dimensions would be $\mathcal{O}(p^2 d^2 N)$, which is more costly. A probabilistic PCA may be used to achieve a comparable complexity of $\mathcal{O}(Dp^2 dNK + dNK^2)$, but is an iterative process that is slower in practice. Tensor factorization methods also pose computational challenges: they require the formulation of a factorization structure for the tensors, and generally have worse time complexity (more discussion in Appendix A.8). Our method can process volumes of size $256^3$ in approximately $\sim 6.5$s per volume with an 11GB RTX 2080 Ti.

**Orthogonal planar reduction.** The reduction hinges on the observation that substantial information from volumes can be preserved given their cross sections from multiple viewpoints (Figure 3). For instance, a feature specific to the left or right side of the brain may be lost from a sagittal profile, but can be recovered when also given the axial profile.

## 4. Experiments

### 4.1. Experimental settings

**Datasets.** We benchmark Raptor on ten datasets that span classification and regression tasks for medical volumes. We summarize the statistics for each dataset in Table A.5.

- **3D Medical MNIST (Yang et al., 2023).** The Medical MNIST (MedMNIST) dataset was proposed as a standardized benchmark for classifying medical imaging modalities. The 3D category of MedMNIST compiles six separate benchmarks based on prior works (Bilic et al., 2023; Armato III et al., 2011; Jin et al., 2020; Yang et al., 2020). Each benchmark presents a classification task, where the number of classes can vary from two to 11. The number of volumes available for training is generally low ($\leq 1.3$k) unlike the 2D datasets in the same benchmark ($\leq 165$k), highlighting the relative data-scarcity of 3D data. The imaging technology underlying the volumes is also diverse, ranging from computed tomography (CT), magnetic resonance imaging (MRI), and electron microscopy (EM).

- **CC-CCII (Zhang et al., 2020).** We explore the China Consortium for Chest CT Image Investigation dataset which presents a classification task between three classes over $2,471$ chest CT scans.

- **CTRG (Tang et al., 2024).** We explore classification tasks over brain MRIs and abdominal CT scans available in the CTRG dataset. A total of $6,000$ brain MRIs were available with 3 multi-class labels (CTRG-B). A total of $1,804$ chest CTs were available with 4 multi-class labels (CTRG-C).

- **UKBB Brain MRIs (Bycroft et al., 2018).** We examine $1,476$ MRIs from the UK Biobank that were available at the time of writing, in conjunction with 162 quantitative

*Table 3.* **3D Medical MNIST volume classification benchmark.** We compare Raptor and state-of-the-art methods on six datasets from the 3D Medical MNIST benchmark. We report AUROC↑ and raw accuracy (ACC↑) (best scores in bold, second best underlined). The methods are grouped by training strategies - from scratch, pretrained on medical datasets, and Raptor.

| | ORGAN | | NODULE | | FRACTURE | | ADRENAL | | VESSEL | | SYNAPSE | |
| METHODS | AUC | ACC | AUC | ACC | AUC | ACC | AUC | ACC | AUC | ACC | AUC | ACC |
|---|---|---|---|---|---|---|---|---|---|---|---|---|
| RESNET | 0.995 | 0.918 | 0.886 | 0.860 | **0.759** | 0.487 | 0.869 | 0.835 | 0.932 | 0.915 | 0.889 | 0.853 |
| MAE | 0.982 | 0.800 | 0.820 | 0.828 | 0.600 | 0.481 | 0.710 | 0.752 | 0.607 | 0.880 | 0.560 | 0.737 |
| MISFM | 0.989 | 0.833 | 0.886 | 0.855 | 0.689 | 0.537 | 0.868 | 0.665 | 0.932 | 0.894 | 0.918 | 0.307 |
| SUPREM | **0.999** | **0.968** | 0.891 | 0.848 | 0.645 | 0.492 | 0.906 | 0.869 | 0.964 | **0.929** | 0.907 | 0.879 |
| SLIVIT | 0.997 | 0.946 | 0.920 | 0.868 | 0.656 | 0.475 | 0.846 | 0.789 | 0.710 | 0.880 | 0.541 | 0.270 |
| VOCO | 0.992 | 0.870 | 0.797 | 0.836 | 0.699 | 0.535 | 0.913 | **0.872** | 0.799 | 0.880 | 0.844 | 0.830 |
| MERLIN | 0.976 | 0.766 | 0.809 | 0.861 | 0.691 | **0.549** | 0.836 | 0.801 | 0.870 | 0.879 | 0.833 | 0.825 |
| RAPTOR-B | 0.998 | 0.958 | 0.904 | 0.858 | 0.647 | 0.501 | **0.930** | 0.858 | 0.945 | 0.919 | 0.922 | 0.894 |
| **RAPTOR** | **0.999** | 0.961 | **0.929** | **0.870** | 0.677 | 0.502 | 0.926 | 0.845 | **0.966** | 0.922 | **0.943** | **0.911** |

imaging-derived phenotypes (IDPs) in the dataset. We categorize IDPs into ten broad categories according to the major regions of the brain and use them as regression targets.

**Baselines.** We evaluate a 3D ResNet-50 trained from scratch, which was a top performer among baselines in the original 3D MedMNIST benchmark (Yang et al., 2023). We additionally fit a 3D ViT on each dataset. To improve the performance of the ViT on the small datasets, we perform masked autoencoder training (MAE) on each training set first, and then fine-tune the model on each downstream task.

**Pretrained models.** We fine-tune several state-of-the-art methods that underwent extensive pretraining for medical volumes, summarized in Table 2. The methods include SLIViT (Avram et al., 2024), SuPreM (Li et al., 2024), Merlin (Blankemeier et al., 2024), MISFM (Wang et al., 2023), and VoCo-L (Wu et al., 2024a). Collectively, our baselines have been demonstrated in prior works to outperform existing pretraining strategies such as SimCLR (Chen et al., 2020), and standard architectures for voxels such as Minkowski (Choy et al., 2019), standard SwinUNETR (Hatamizadeh et al., 2021) and UniMiSS (Xie et al., 2024). For models which do not have a classification or regression head, we add a single linear layer above their latent space to enable the downstream tasks.

To perform downstream tasks using Raptor embeddings, we train logistic regression models with L2 regularization (tuning penalties of 0.01, 0.1, 1.0, 10.0, and 100.0 based on the validation set) for classification tasks[4], and MLPs (tuned up to 3 layers based on the validation set) under an MSE loss (described further in Appendix A.2) for regression tasks. This workflow is visualized in Figure 4.

**Evaluation method.** We measure AUROC (AUC) and accuracy (ACC) to benchmark all classification datasets

---

[4]Logistic Regression in scikit-learn

*Table 4.* **Classification results on the CC-CCII, CTRG-C, and CTRG-B datasets.** We compare Raptor and state-of-the-art methods on other publicly available medical volume datasets with annotations. We report AUROC↑ and raw accuracy (ACC↑) (best scores in bold, second best underlined - scoring follows MedMNIST standard).

| | CC-CCII | | CTRG-C | | CTRG-B | |
| METHODS | AUC | ACC | AUC | ACC | AUC | ACC |
|---|---|---|---|---|---|---|
| RESNET | 0.933 | 0.792 | 0.580 | 0.721 | 0.592 | 0.791 |
| MAE | 0.937 | 0.813 | 0.529 | 0.275 | 0.601 | 0.800 |
| MISFM | 0.975 | 0.835 | 0.548 | 0.690 | 0.651 | 0.643 |
| SUPREM | 0.988 | 0.940 | 0.613 | 0.759 | **0.717** | 0.830 |
| SLIVIT | 0.986 | 0.906 | 0.571 | 0.719 | 0.656 | 0.834 |
| VOCO | 0.881 | 0.733 | 0.526 | 0.750 | 0.636 | 0.802 |
| MERLIN | 0.933 | 0.815 | 0.567 | 0.750 | 0.591 | 0.833 |
| RAPTOR-B | 0.992 | 0.954 | 0.600 | 0.754 | 0.685 | **0.842** |
| **RAPTOR** | **0.997** | **0.955** | **0.620** | **0.767** | 0.711 | **0.842** |

(consistent with Yang et al. 2023), and use Pearson's $r^2$ for all regression datasets. For all methods, we reserve the validation set of the datasets for hyperparameter tuning, and report all final accuracies on the test set. For the 3D Medical MNIST dataset, we use the predetermined data splits from the benchmarks, allowing a direct comparison with prior and future works. For all other datasets without the standard splits, we use random training, validation, test splits of 60%/20%/20%.

### 4.2. Main results: Classification

Raptor achieves strong performance across classification tasks (3D MedMNIST, CC-CCII, and CTRG; Tables 3 & 4), evaluated using AUROC and accuracy as in the MedMNIST benchmark (Yang et al., 2023). It obtains the highest accuracy in 6 of 9 datasets and ranks among the top two on 8 of 9 (by AUC). When combined, Raptor and Raptor-B achieve the best–reported AUCs on five 3D MedMNIST categories.

*Table 5.* **UKBB Brain IDP regression benchmark.** We compare Raptor with state-of-the-art methods on predicting quantitative traits derived from ten different groups of regions from the brain (best scores in bold, second best underlined).

| METHODS | WHITEM | GRAYM | CEREB | AMYG | HIPPO | CORTEX $r^2$ | GYRUS | PALL | CAUD | THAL | AVG. |
|---|---|---|---|---|---|---|---|---|---|---|---|
| RESNET | 0.417 | 0.562 | 0.193 | 0.072 | 0.108 | 0.125 | 0.099 | 0.055 | 0.162 | 0.134 | 0.193 |
| MAE | 0.036 | 0.045 | 0.072 | 0.036 | 0.040 | 0.043 | 0.032 | 0.012 | 0.037 | 0.036 | 0.039 |
| MISFM | 0.418 | 0.624 | 0.276 | 0.089 | 0.145 | 0.236 | 0.209 | 0.087 | 0.166 | 0.164 | 0.242 |
| SUPREM | 0.646 | 0.696 | 0.330 | 0.109 | 0.163 | 0.275 | 0.256 | 0.067 | 0.255 | 0.195 | 0.299 |
| SLIVIT | 0.474 | 0.694 | 0.258 | 0.134 | 0.190 | 0.268 | 0.213 | 0.053 | 0.192 | 0.174 | 0.265 |
| VOCO | 0.225 | 0.375 | 0.189 | 0.071 | 0.113 | 0.059 | 0.048 | 0.043 | 0.060 | 0.075 | 0.126 |
| MERLIN | 0.622 | 0.734 | 0.335 | 0.127 | 0.180 | 0.313 | 0.269 | 0.093 | 0.247 | 0.210 | 0.313 |
| RAPTOR-B | 0.614 | 0.742 | 0.398 | **0.185** | 0.247 | 0.355 | 0.314 | 0.116 | 0.331 | 0.258 | 0.356 |
| **RAPTOR** | **0.681** | **0.777** | **0.437** | 0.170 | **0.262** | **0.404** | **0.340** | **0.142** | **0.381** | **0.300** | **0.389** |

Models that leverage pretraining (MISFM, SuPreM, SLIViT, VoCo, Merlin, Raptor) generally score higher than those trained from scratch (3D ResNet and MAE). Across all classification datasets, Raptor on average improved metrics by $+2\%$ over SuPreM, $+4\%$ over MISFM, $+9\%$ over Merlin, $+10\%$ over VoCo, and $+13\%$ over SLIViT.

### 4.3. Main results: Regression

We evaluate regression performance using $r^2$ across 10 brain regions from UK Biobank Brain MRI volumes and their associated IDPs (Table 5). We observe that Raptor shows strong predictive accuracy on volumetric traits, achieving the highest $r^2$ scores in 9 out of 10 regions. Raptor-B ranks second overall, outperforming Raptor in one region (Amygdala). Merlin and SuPreM follow as the next-best performers. Interestingly, the ResNet baseline bests several pretrained models, suggesting that current volumetric pretraining strategies may not generalize well between imaging modalities such as CT and MRI. Models generally perform better on broader volumetric measures (e.g., white matter, gray matter) than on more localized regions (e.g., Hippocampus). Compared to other methods, Raptor improves average performance by $+24\%$ over Merlin, $+30\%$ over SuPreM, $+47\%$ over SLIViT, $+61\%$ over MISFM, and over $+100\%$ relative to the remaining baselines. Full performance comparisons are visualized in Figure 1, with additional metrics provided in Appendix A.10.

### 4.4. Effectiveness in data-constrained settings

We observe that Raptor can generalize effectively to downstream tasks that have a limited number of samples – a common scenario in the medical domain (Figure 5). We measure the AUROC of predictions using a single linear layer on top of Raptor embeddings given a limited number of samples $(10, 50, 100, 200, 500)$ in the 3D MedMNIST Synapse dataset, which originally contains $1,230$ samples for training. Even with 10 samples, Raptor obtains 77%

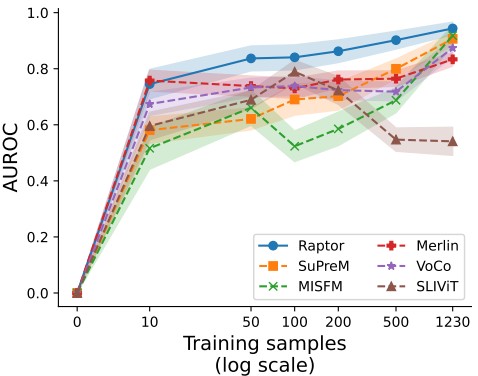

*Figure 5.* **Prediction accuracy with limited training data.** We measure the effect of limiting the number of training samples between $10 \sim 500$ on the final test accuracy on the Synapse dataset. Shaded area represents 95% CI.

$(0.729)$ of the performance achieved with the full $1,230$ samples $(0.943)$. With 100 samples, Raptor obtains 88% $(0.838)$ of the full-sample performance. Results for additional datasets show similar trends (Figure A.2).

### 4.5. Efficiency of the embeddings

We find that Raptor-B demonstrates exceptional capabilities despite its representation being 90% smaller than Raptor and 97% smaller than that of SuPreM (next-best, Figure 6). In classification, Raptor-B matches SuPreM in average accuracy, and surpasses it in regression. We also highlight the efficiency of Raptor with 100 projections – it consistently outperforms all baselines on average while being smaller, including the feature map inferred by VoCo $(3072 \times 3^3)$. Our results highlight the potential of Raptor for compressing and sharing medical volumes, with the flexibility to adjust $K$ as needed. As demonstrated in Section 5.1, Raptor achieves state-of-the-art performance, while Raptor-B offers

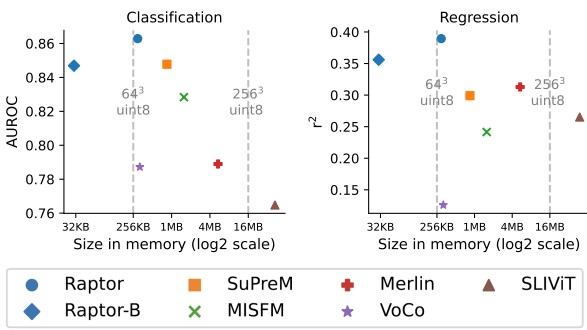

Figure 6. **Overall accuracy of each method in the context of their embedding sizes.** Average performance of each method on the classification and regression datasets according to the size of the latent space inferred by each method (single precision floats).

substantial space savings with only a marginal decrease in performance.

## 5. Ablation Studies and Analyses

We quantitatively validate our embedding strategy by testing it against alternative approaches. For all ablations, we report the average score on the 3D MedMNIST dataset to assess their impact.

### 5.1. Reliability of random projections

We evaluate the sensitivity of our approach to the number of random projections used to compress the DINOv2-L latent representations. Specifically, we test $K$ values of $1, 5, 10, 100$ and $150$ across three different random seeds (Table 6). As expected, smaller $K$ results in lower accuracy and higher variance across seeds. However, performance plateaus after $K = 100$, with negligible variation between seeds (standard deviation $< 0.001$), indicating that random projections yield stable embeddings at each run. These results suggest that the embedding size can be substantially reduced with only a minor tradeoff in stability, making the approach well-suited for resource-constrained settings. In Table A.3, we further show the robustness of Raptor across a larger range of seeds (integers from 0 to 9), showing consistent performance in up to 10 independent runs.

### 5.2. Using multiple viewpoints

To assess the importance of multi-view processing, we compare models trained on fewer views (one or two) instead of all three (Table 7). We find that incorporating multiple views always leads to better classification. The largest performance gain is observed when increasing from one to two views, with a smaller but positive gain when using all three. This confirms the benefit of aggregating information from multiple viewpoints when parsing 3D volumes.

Table 6. **Reliability of random projections.** We vary the number of random projections for Raptor embeddings, and examine the variance in the average AUC on the 3D MedMNIST dataset.

| METHOD | K | SEED 1 | SEED 2 | SEED 3 | STD. |
|---|---|---|---|---|---|
| **RAPTOR** | 1 | 0.818 | 0.817 | 0.793 | 0.0116 |
| | 5 | 0.890 | 0.860 | 0.864 | 0.0133 |
| | 10 | 0.866 | 0.896 | 0.876 | 0.0127 |
| | 100 | 0.901 | 0.899 | 0.900 | 0.0008 |
| | 150 | 0.898 | 0.897 | 0.897 | 0.0004 |

Table 7. **Effect of using fewer views of volumes.** We vary which orthogonal views to parse when computing Raptor embeddings, and compare their average performance on the 3D MedMNIST dataset.

| METHOD | VIEWPOINT | AUC | ACC |
|---|---|---|---|
| **RAPTOR** | (A)XIAL | 0.887 | 0.780 |
| | (C)ORONAL | 0.862 | 0.814 |
| | (S)AGITTAL | 0.881 | 0.806 |
| | A,C | 0.893 | 0.826 |
| | C,S | 0.900 | 0.812 |
| | A,S | 0.884 | 0.822 |
| | A,C,S | 0.901 | 0.838 |

### 5.3. Limitation on detectable features

To explore the spatial sensitivity and resolution limits of our method, we conduct two controlled simulations using MNIST digits inserted into medical volumes from the 3D MedMNIST Nodule dataset.

In the **Location task**, we embed the same MNIST digit into two volumes, with varying lateral pixel (px) distances apart ($64$px to $8$px), and test whether Raptor can distinguish the two.

In the **Size task**, we randomly insert an MNIST digit of varying sizes (from $64$px to $8$px) into volumes from the 3D MedMNIST dataset, and test whether Raptor can detect the presence of the digit.

For both tasks, we train binary classifiers on top of the Raptor embeddings. As expected, classification becomes increasingly difficult as spatial resolution decreases (Table 8).

In the location task, Raptor achieves AUC $\sim 0.8$ even at $8$px resolution, highlighting the high spatial specificity of our embeddings. In the size task, performance is near perfect at the largest scale ($> 0.9$ AUC and ACC for $64$px), but drops sharply for smaller digits (e.g., AUC $\sim 0.5$ at $16$px), reflecting the limits of detection at low spatial scales. We further explore whether the choice of digit can lead to variability in the simulation results in Table A.4.

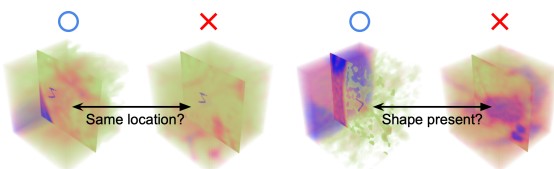

*Figure 7.* **Examples of the "location" and "size" detection simulated tasks.** The "location" task asks whether a digit appears in an expected location in a volume. The "size" task asks whether a digit of varying sizes is present in a volume.

*Table 8.* **Results of "location" and "size" simulated benchmarks.** We perform simulations that place MNIST digits at varying locations and sizes in 3D MedMNIST volumes, and pose binary classification tasks to detect the digits with the predetermined settings.

| METHOD | TASK | RESOLUTION | AUC | ACC |
|--------|------|-----------|------|------|
| **RAPTOR** | LOC. | 64PX | 0.941 | 0.886 |
| | | 32PX | 0.892 | 0.822 |
| | | 16PX | 0.912 | 0.855 |
| | | 8PX | 0.783 | 0.701 |
| **RAPTOR** | SIZE | 64PX | 0.988 | 0.932 |
| | | 32PX | 0.758 | 0.682 |
| | | 16PX | 0.520 | 0.507 |
| | | 8PX | 0.474 | 0.476 |

## 6. Discussion

Raptor achieves state-of-the-art performance across a range of medical imaging tasks, while compressing volumes into a representation space that is significantly smaller than those used in prior work. On average, Raptor outperforms all competing methods, achieving a 3% accuracy gain over the next best approach (SuPreM), despite using an embedding that is 2.9× smaller. With a further ten-fold compression, Raptor-B matches SuPreM's performance while reducing embedding size by 28.8×. Our results underscore the efficacy of Raptor as a paradigm-shifting framework for volumetric data analysis that overcomes major challenges in the field by leveraging pretrained 2D foundation models.

Raptor is an original approach for bridging the gap between widely studied 2D foundation models and 3D volumetric tasks in a domain-agnostic manner. Unlike existing methods that rely on task-specific pretraining (e.g., VoCo, Merlin) or complex architectural modifications, our approach bypasses these inefficiencies through a projection- and reduction-based pipeline. A key strength of Raptor lies in its flexibility: the 2D foundational model it relies on can be seamlessly upgraded as better models emerge.

This work provides both theoretical and empirical justifications for our design choices, rather than relying on heuristics or purely practical considerations. We position Raptor as a scalable and generalizable solution for resource-constrained settings, particularly in domains where annotated volumetric datasets are limited and computational efficiency is critical. Extending our framework beyond healthcare applications is a promising direction to explore in future work.

Despite these advancements, certain limitations remain. For instance, modest performance on select datasets (e.g. Fracture3D) suggests that domain-specific priors or refining the axial sampling strategy could further improve downstream results. In supplementary analysis (Figure A.3), we identify contributing factors to such performance drops, pointing to immediate next steps for improvement. Nonetheless, the results establish Raptor as a novel and impactful contribution to the intersection of foundation models and dense volumetric learning, offering a computationally efficient alternative to traditional 3D learning paradigms. These findings hold promise for broader applications, including multimodal integration and large-scale analysis of medical data.

## Impact Statement

We introduce Raptor, a paradigm-changing framework that dramatically reduces the computational and data requirements for high-dimensional medical volume analysis. By combining pretrained 2D foundation models with efficient random projection techniques, Raptor makes state-of-the-art volumetric analysis more accessible to researchers with limited computational resources. Our method not only improves performance across a wide range of medical imaging tasks but also enables scalable and cost-effective deployment in data-scarce or resource-constrained settings. By lowering technical and computational barriers, Raptor has the potential to broaden participation in machine learning research and drive innovation in volumetric data analysis across healthcare and beyond.

**Acknowledgments**  We thank Eric Liu, Jeffrey Chiang, Richard Border, and Oren Avram for their feedback and discussions throughout this project. This research was conducted using the UK Biobank Resource under application 331277. We thank the participants of UK Biobank for making this work possible. This research was supported by grants from the National Institutes of Health (NIH) and the National Science Foundation (NSF). The work was supported by NIH grant R35GM153406 and NSF grant CAREER-1943497. SL is supported by the Warren Alpert Fellowship. AG was supported in part by NIH grant R01MH130581 and R01MH122569.

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

# A. Appendix

## A.1. Using alternate image foundation models in Raptor

*Table A.1.* Effectiveness of other 2D ViT encoders

| METHOD | ENCODER | AUC | ACC |
|---|---|---|---|
| **RAPTOR** | SAM | 0.850 | 0.800 |
| | CLIP | 0.870 | 0.813 |
| | LLAVA-MED | 0.869 | 0.812 |
| | MEDSAM | 0.872 | 0.808 |
| | DINOV2-L | 0.907 | 0.835 |

As our approach is agnostic to the choice of the image foundation model, we explore CLIP (Radford et al. 2021; also utilized in Moor et al. 2023), the encoder of SAM (Kirillov et al., 2023), the encoder of Llava-Med (Li et al., 2023), and the encoder of MedSAM (Ma et al., 2024). Of the explored options, embeddings obtained using DINOv2 lead to the highest accuracies in 3D MedMNIST (Table A.1). We also find that embeddings based on the MedSAM encoder do not lead to significantly higher accuracies than those based on SAM, despite MedSAM being directly trained on medical imaging modalities. Overall, we note that our findings are roughly in line with El Banani et al. (2024) and Oquab et al. (2023), which determined that DINOv2 currently yields the most informative features for general-purpose downstream tasks.

## A.2. Multi-Layer Perceptron for downstream

We train a multi-layer perceptron (MLP) to predict brain imaging-derived phenotypes (IDPs) in the UK Biobank. The configuration of the MLP is described below for the case of K=100 Raptor embeddings. We use PyTorch to implement and train the MLP module. Training proceeded up to 50 epochs using the mean-squared error loss. The model weights are checkpointed only when accuracy improved on the validation split of each dataset.

```
MLP(
  (compare_bn): BatchNorm1d(199)
  (model): Sequential(
    (0): Linear(77100, 256)
    (1): BatchNorm1d(256)
    (2): ReLU()
    (3): Linear(256, 256)
    (4): ReLU()
    (5): Linear(256, 162)
  )
)
```

*Listing 1.* 3-layer multi-layer perceptron

## A.3. Quantifying how much medical features can be captured using DINOv2-L

We explore one way of quantifying how effectively DINOv2-L extracts features related to the medical domain. The high-level intuition is that the semantically meaningful features of medical volumes are still part of the universal image features (edges, textures, shapes), which a broad 2D encoder has already learned from large-scale, diverse data. Raptor uses generic image foundation models like DINOv2-L to capture any meaningful image features and performs low rank approximations to retain only the relevant ones. This is in contrast to many existing medical image/volume models that try to learn such mappings from scratch or fine-tune a general model to be biased towards them.

To illustrate this point, we projected both generic (ImageNet) and medical (3D MedMNIST) embeddings onto the principal components derived from the generic dataset. In the table below, we calculate the total variance explained as we add more PCs (Table A.2). Although the medical data do not align with the very top principal directions, the variance of the medical dataset increases as more components are added (eventually matching the generic variance), demonstrating that medical image features lie within the general image feature space.

*Table A.2.* **Effectiveness of DINOv2-L in the medical domain.** Here we compare how much of the variance is explained for the two datasets' DINOv2-L embeddings, as we add more principal components (PCs) from the generic image embeddings. As can be seen below, with more generic image embedding PCs, variance in the medical dataset embeddings can be explained in near full.

| #PCs | General Explained Var. | Medical Explained Var. | Ratio (Medical/General) |
|------|------------------------|------------------------|--------------------------|
| 100  | 0.613                  | 0.225                  | 0.367                    |
| 200  | 0.765                  | 0.397                  | 0.518                    |
| 500  | 0.926                  | 0.734                  | 0.793                    |
| 1000 | 0.999                  | 0.994                  | 0.996                    |

*Table A.3.* **Reliability of random projections.** We vary the number of random projections for Raptor embeddings, and examine the variance in the average AUC on the MedMNIST dataset.

| METHOD | K | SEED 1 | 2 | 3 | 4 | 5 | 6 | 7 | 8 | 9 | 10 | AVG. | STD. |
|--------|-----|--------|-------|-------|-------|-------|-------|-------|-------|-------|-------|-------|--------|
| **RAPTOR** | 1   | 0.818 | 0.817 | 0.793 | 0.823 | 0.831 | 0.818 | 0.831 | 0.814 | 0.801 | 0.831 | 0.818 | 0.0121 |
|        | 5   | 0.890 | 0.860 | 0.864 | 0.869 | 0.869 | 0.877 | 0.868 | 0.858 | 0.872 | 0.866 | 0.869 | 0.0086 |
|        | 10  | 0.866 | 0.896 | 0.876 | 0.875 | 0.882 | 0.879 | 0.871 | 0.876 | 0.877 | 0.886 | 0.878 | 0.0078 |
|        | 100 | 0.901 | 0.899 | 0.900 | 0.898 | 0.905 | 0.891 | 0.899 | 0.898 | 0.900 | 0.898 | 0.899 | 0.0034 |
|        | 150 | 0.898 | 0.897 | 0.897 | 0.894 | 0.898 | 0.903 | 0.902 | 0.905 | 0.902 | 0.904 | 0.900 | 0.0034 |

## A.4. MRI Anatomical Terminology

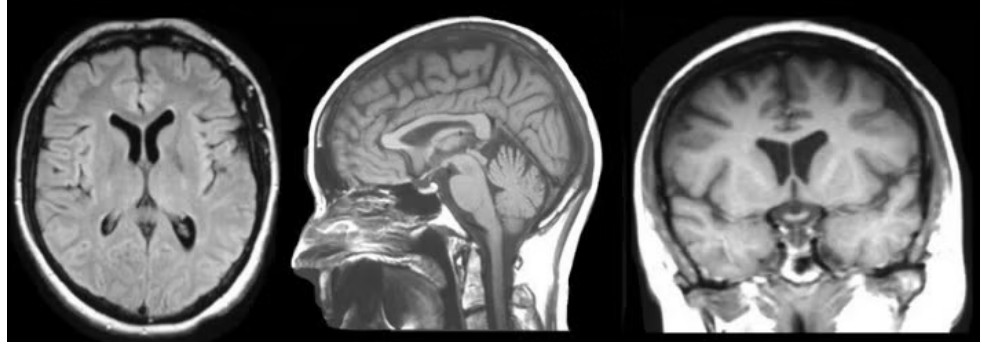

*Figure A.1.* **Brain Anatomy.** Axial, Sagittal, and Coronal views of a brain MRI volume (source: https://case.edu/med/neurology)

*Table A.4.* **Variability in simulated settings under specific digits.** We repeat the "Size" simulation experiment given specific digits (0, 1, and 8) found in the MNIST dataset.

| Method | Size | 0 | 1 | 8 | Avg. | Std. |
|--------|------|-------|-------|-------|-------|-------|
| **Raptor** | 64px | 0.798 | 0.678 | 0.753 | 0.743 | 0.049 |
|        | 32px | 0.580 | 0.559 | 0.575 | 0.571 | 0.009 |
|        | 16px | 0.552 | 0.514 | 0.517 | 0.527 | 0.017 |
|        | 8px  | 0.506 | 0.509 | 0.508 | 0.508 | 0.001 |

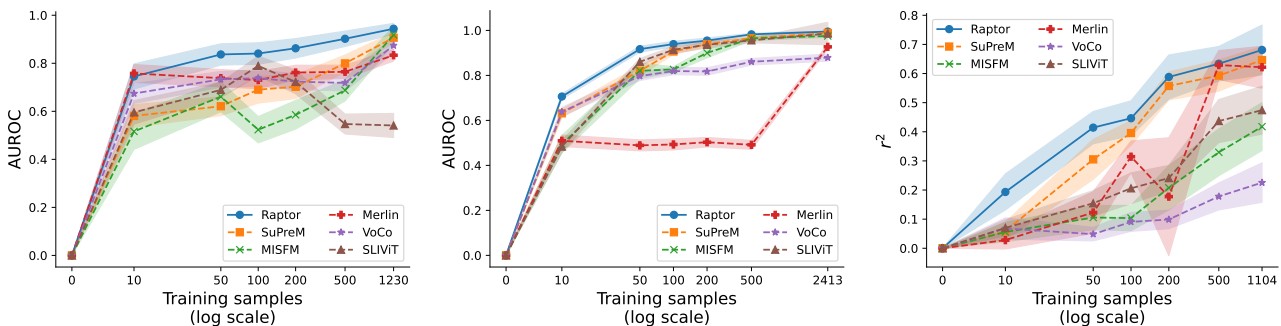

*Figure A.2.* **Prediction accuracy with limited training data.** We measure the effect of limiting the number of training samples between $10 \sim 500$ on the final test accuracy on the Synapse, CC-CCII, and UKBB (White Matter - regression) datasets. Shaded area represents 95% CI.

## A.5. Formal Guarantees on Random Projections

To give a more principled view of our random projection step, we restate a version of the Johnson–Lindenstrauss (JL) lemma (Dasgupta & Gupta, 2003) and show how it applies in our multi-view volumetric setting.

**Full Proof of the Johnson–Lindenstrauss Lemma**    Let $\mathbf{x}_1, \mathbf{x}_2, \ldots, \mathbf{x}_n \in \mathbb{R}^d$ be arbitrary vectors, and fix $0 < \varepsilon < 1$. Let $\mathbf{R} \in \mathbb{R}^{K \times d}$ be a random projection matrix whose entries are drawn i.i.d. from a sub-Gaussian distribution (e.g., Gaussian) with zero mean and variance $\sigma^2$ chosen so that $\mathbb{E}\big[\|\mathbf{R}\mathbf{z}\|_2^2\big] = \|\mathbf{z}\|_2^2$ for any $\mathbf{z} \in \mathbb{R}^d$. We show that if $K$ is on the order of $\varepsilon^{-2} \log(n)$, then with high probability for all pairs $i \neq j$,

$$(1 - \varepsilon) \, \|\mathbf{x}_i - \mathbf{x}_j\|_2^2 \ \leq \ \|\mathbf{R}\,\mathbf{x}_i - \mathbf{R}\,\mathbf{x}_j\|_2^2 \ \leq \ (1 + \varepsilon) \, \|\mathbf{x}_i - \mathbf{x}_j\|_2^2.$$

**I: *Single-Pair Analysis.*** Consider a single pair $(i, j)$ and let $\mathbf{z} = \mathbf{x}_i - \mathbf{x}_j$. The vector $\mathbf{R}\mathbf{z}$ has i.i.d. sub-Gaussian components with mean 0 and variance on the order of $\|\mathbf{z}\|_2^2 / K$. In particular, if $\mathbf{R}$ has Gaussian entries $\sim \mathcal{N}(0, 1/K)$, then each $\langle \mathbf{r}_\ell, \mathbf{z} \rangle$ is $\mathcal{N}(0, \|\mathbf{z}\|_2^2 / K)$ and $\|\mathbf{R}\mathbf{z}\|_2^2$ follows a scaled $\chi^2(K)$ distribution with expected value $\|\mathbf{z}\|_2^2$.

**II: *Concentration Inequality.*** Sub-Gaussian concentration (or standard $\chi^2$ tail bounds) implies that for any $\varepsilon > 0$, there exist positive constants $c_1, c_2$ so that

$$\Pr\Big[\big|\,\|\mathbf{R}\mathbf{z}\|_2^2 - \|\mathbf{z}\|_2^2\big| \ \geq \ \varepsilon \, \|\mathbf{z}\|_2^2\Big] \ \leq \ c_1 \, \exp\!\big(-c_2 \, \varepsilon^2 \, K\big).$$

Hence, $\|\mathbf{R}\mathbf{z}\|_2^2$ is within $(1 \pm \varepsilon)\|\mathbf{z}\|_2^2$ with high probability for one pair $(i, j)$.

**III: *Union Bound over All Pairs.*** There are $\binom{n}{2} \leq n^2/2$ distinct pairs $(i, j)$. A union bound over these pairs requires

$$n^2 \, c_1 \, \exp\!\big(-c_2 \, \varepsilon^2 \, K\big) \ \leq \ 1,$$

which rearranges to

$$K \ \geq \ C \, \varepsilon^{-2} \, \log(n)$$

for some constant $C$. This guarantees that with high probability, for all $i \neq j$,

$$(1 - \varepsilon) \, \|\mathbf{x}_i - \mathbf{x}_j\|_2^2 \ \leq \ \|\mathbf{R}\,\mathbf{x}_i - \mathbf{R}\,\mathbf{x}_j\|_2^2 \ \leq \ (1 + \varepsilon) \, \|\mathbf{x}_i - \mathbf{x}_j\|_2^2.$$

This completes the proof that random projections preserve pairwise distances among $\{\mathbf{x}_1, \ldots, \mathbf{x}_n\}$ up to $(1 \pm \varepsilon)$ distortion.

## A.6. Mathematical formulation of Raptor

Assume $\boldsymbol{x} \in \mathbb{R}^{D \times D \times D}$ be a 3D volumetric scan. For each planar direction (defined along the axes $x, y, z$), we get $D$ slices $S_x = \{\boldsymbol{s}_{x_1}, \boldsymbol{s}_{x_2}, \ldots, \boldsymbol{s}_{x_D}\}, S_y, S_z$. We use a pretrained ViT (e.g., DINOv2-L) encoder that takes each slice $\boldsymbol{s}_{ij}$ to generate a feature vector $\boldsymbol{z}_{ij}$:

$$\phi(\cdot) : \mathbb{R}^{D \times D} \to \mathbb{R}^m$$

In practice, the encoder outputs a token-wise embedding for a grid of patches, but we may flatten it to a $m$ dimensional vector ($m = dp^2$, where $d$ is the latent dimension of token embedding, $p^2$ is the number of patches). These feature vectors are *semantically rich*, meaning they encode information that is reflective of real-world semantics, such as contours of similar objects, distributions of certain features, etc. Ideally we would like to leverage all embeddings of the slices for all three planar views; however this is computationally prohibitive:

$$(\phi(S_x), \phi(S_y), \phi(S_z)) \in \mathbb{R}^{3D \times dp^2}$$

Instead, Raptor aggregates the slice-level token embeddings and low-rank approximates them for each planar view (note that due to the linearity of random projection and mean aggregation, the two operations can be done in any order). Reducing the planar view by mean-pooling the slices gives

$$\bar{\boldsymbol{z}}_i = \frac{1}{D} \sum_{j=1}^{D} \boldsymbol{z}_{ij}, \quad \text{where } i \in \{x, y, z\}$$

Define $\boldsymbol{R} \in \mathbb{R}^{K \times d}$ a Random Projection matrix, whose entries are sampled from the standard normal.

$$\boldsymbol{u}_i = \boldsymbol{R}\bar{\boldsymbol{z}}_i \in \mathbb{R}^{Kp^2}, \quad K \ll d, \quad \boldsymbol{R}_{kl} \sim \mathcal{N}(0, 1)$$

Concatenating the three low-rank approximated planar embeddings gives the final Raptor embedding:

$$\Phi(\boldsymbol{x}) = \text{concat}\,[\boldsymbol{u}_x, \boldsymbol{u}_y, \boldsymbol{u}_z] \in \mathbb{R}^{3Kp^2}$$

Or alternatively:

$$\Phi(\boldsymbol{x}) = \text{concat}\left[ \boldsymbol{R}\frac{1}{D}\sum_{j=1}^{D}\phi(\boldsymbol{s}_{x_j}), \boldsymbol{R}\frac{1}{D}\sum_{j=1}^{D}\phi(\boldsymbol{s}_{y_j}), \boldsymbol{R}\frac{1}{D}\sum_{j=1}^{D}\phi(\boldsymbol{s}_{z_j}) \right]$$

Because (with the exception of $\phi(\cdot)$) all our mappings are linear, we can analyze the pairwise distances in different scales: (1) the pairwise distances between two volumes, and (2) the pairwise distances between two classes (in a downstream task of classification).

First, we formulate how Raptor is able to preserve the difference in the semantic embeddings of two distinct volumes in the three orthogonal axes. Let $\boldsymbol{x_a}, \boldsymbol{x_b}$ be two 3D volumetric scans. Their high-dimensional semantic embeddings are given as:

$$\phi(\boldsymbol{x_a}) = \{\phi(S_x^a), \phi(S_y^a), \phi(S_z^a)\}, \quad \phi(\boldsymbol{x_b}) = \{\phi(S_x^b), \phi(S_y^b), \phi(S_z^b)\}, \quad \phi(\boldsymbol{x_a}), \phi(\boldsymbol{x_b}) \in \mathbb{R}^{3 \times D \times dp^2}$$

For each axial view, the difference is given as (taking the Frobenius norm)

$$\|\phi(\boldsymbol{x_a})_x - \phi(\boldsymbol{x_b})_x\|_F = \left\|\phi(S_x^a) - \phi(S_x^b)\right\|_F$$

$$= \sqrt{\sum_{j=1}^{D}\left[\phi(\boldsymbol{s}_{x_j}^a) - \phi(\boldsymbol{s}_{x_j}^b)\right]^2}$$

The Raptor embeddings of the two volumes are given as:

$$\Phi(\boldsymbol{x_a}) = \text{concat}\left[ \frac{1}{D}\sum_{j=1}^{D}\boldsymbol{R}\phi(\boldsymbol{s}_{x_j}^a), \frac{1}{D}\sum_{j=1}^{D}\boldsymbol{R}\phi(\boldsymbol{s}_{y_j}^a), \frac{1}{D}\sum_{j=1}^{D}\boldsymbol{R}\phi(\boldsymbol{s}_{z_j}^a) \right] \in \mathbb{R}^{3kp^2}$$

$$\Phi(\boldsymbol{x_b}) = \text{concat}\left[ \frac{1}{D}\sum_{j=1}^{D}\boldsymbol{R}\phi(\boldsymbol{s}_{x_j}^b), \frac{1}{D}\sum_{j=1}^{D}\boldsymbol{R}\phi(\boldsymbol{s}_{y_j}^b), \frac{1}{D}\sum_{j=1}^{D}\boldsymbol{R}\phi(\boldsymbol{s}_{z_j}^b) \right] \in \mathbb{R}^{3kp^2}$$

and the distance between the two along the $x$ axis is:

$$\|\Phi(\boldsymbol{x}_a)_x - \Phi(\boldsymbol{x}_b)_x\| = \frac{1}{D} \left\| \sum_{j=1}^{D} \boldsymbol{R}\phi(\boldsymbol{s}_{x_j}^a) - \sum_{j=1}^{D} \boldsymbol{R}\phi(\boldsymbol{s}_{x_j}^b) \right\|$$

$$= \frac{1}{D} \left\| \sum_{j=1}^{D} \boldsymbol{R}\left( \phi(\boldsymbol{s}_{x_j}^a) - \phi(\boldsymbol{s}_{x_j}^b) \right) \right\|$$

Define $\Delta_j = \phi(\boldsymbol{s}_{x_j}^a) - \phi(\boldsymbol{s}_{x_j}^b)$ as the difference in the slice-wise DINOv2-L embeddings. Then we may rewrite the distances in the raw embeddings and Raptor embeddings for one of the axial views as

$$d_{raw} = \sqrt{\sum_{j=1}^{D} \Delta_j^2}, \quad d_{Raptor} = \frac{1}{D} \left\| \sum_{j=1}^{D} \boldsymbol{R}\Delta_j \right\|$$

If $\Delta_j$ are aligned in the same direction, then there exists a trivial mapping between $d_{raw}$ and $d_{Raptor}$, but if there are mis-alignments (i.e., $\Delta_j$ and $\Delta_{j'}$ are in opposite directions), then that is no longer the case. Given that we are working with 3D volumetric scans of real objects that have certain properties, we make an additional assumption that the slices are "smooth". Because DINOv2-L is a (locally) Lipschitz mapping, smoothness in the raw volumes also mean the embeddings of the slices are smooth. Assume that for some $\alpha_j > 0$ values (again taking the $l_2$ norms in flattened form or Frobenius norm in matrix form), the following holds:

$$\Delta_j^\top S_{j-1} \geq \alpha_j \|\Delta_j\| \|S_{j-1}\|, \quad j = 1, \ldots, D$$

Where $S_j = \sum_{k=1}^{k=j} \Delta_k$ is the partial sum of the distances. Geometrically, this means that at each slice level, $\Delta_j$ deviates from $S_{j-1}$ by at most $\arccos(\alpha_j)$. This ensures that the slices (and their embeddings due to the Lipschitz bound) are globally smooth, and there are no abrupt (negative) changes in the differences at each slice level. Now we derive the bounds for $d_{Raptor}$ by induction. In the base case we have:

$$S_1 = \Delta_1, \quad \|S_j\| = \|\Delta_j\|$$

For the $j$-th slice,

$$S_{j+1} = S_j + \Delta_{j+1}, \quad \|S_{j+1}\|^2 = \|S_j\|^2 + \|\Delta_{j+1}\|^2 + 2S_j^\top \Delta_{j+1}$$

$$\geq \|S_j\|^2 + \|\Delta_{j+1}\|^2 + 2\alpha_j \|S_j\| \|\Delta_{j+1}\|$$

$$\geq (\|S_j\| + \alpha_j \|\Delta_{j+1}\|)^2 \qquad \text{(since } 0 < \alpha_j \leq 1)$$

$$\Rightarrow \|S_{j+1}\| \geq \|S_j\| + \alpha_j \|\Delta_{j+1}\|$$

Which we can unroll to the $j = D$ slice:

$$\|S_D\| = \left\| \sum_{j=1}^{D} \Delta_j \right\| \geq \sum_{j=1}^{D} \alpha_j \|\Delta_j\|$$

$$\geq \sqrt{\sum_{j=1}^{D} \alpha_j^2 \|\Delta_j\|^2} \qquad \text{(since } \alpha_j \|\Delta_j\| \geq 0)$$

$$\geq \alpha_{min} \sqrt{\sum_{j=1}^{D} \|\Delta_j\|^2}, \quad \alpha_j \geq \alpha_{min}$$

Giving us a lower bound in aggregating the $\Delta_j$:

$$\left\| \sum_{j=1}^{D} \Delta_j \right\| \geq \alpha_{min} \left\| \phi(S_x^a) - \phi(S_x^b) \right\|_F = \alpha_{min} d_{raw}$$

Applying random projection gives us the distance in the Raptor embeddings of the two volumes. Therefore we get the final lower bound in the distances:

$$\|\Phi(\boldsymbol{x}_a)_x - \Phi(\boldsymbol{x}_b)_x\| = \frac{1}{D}\left\|\boldsymbol{R}\left(\sum_{j=1}^{D}\Delta_j\right)\right\| \geq \frac{(1-\varepsilon)}{D}\alpha_{min}\sqrt{\sum_{j=1}^{D}\|\Delta_j\|^2} = \frac{(1-\varepsilon)}{D}\alpha_{min}d_{raw}$$

Similarly, we can derive the upper bound of the distance in the Raptor embeddings.

$$\left\|\sum_{j=1}^{D}\Delta_j\right\| \leq \sum_{j=1}^{D}\|\Delta_j\| \qquad\qquad \text{(triangle inequality)}$$

$$\leq \sqrt{D}\cdot\sqrt{\sum_{j=1}^{D}\|\Delta_j\|^2} \qquad\qquad \text{(Cauchy-Schwartz inequality)}$$

Giving a similar bound as before with respect to the raw Raptor embedding distance. The final bounds on both ends are:

$$\frac{(1-\varepsilon)}{D}\alpha_{min}d_{raw} \leq d_{Raptor} \leq \frac{(1+\varepsilon)}{\sqrt{D}}d_{raw}$$

In short, under the assumption that $\alpha_j > 0$, the Raptor difference does not vanish unless there is no difference in the raw embeddings of all the slices between two volumes, nor does it blow up if there is a large distinction in the original embedding space. This means that if there are any differences between the two volumes in the semantically-rich DINOv2-L space, these differences will be preserved in our embeddings. The presence of upper- and lower-bounds of $d_{Raptor}$ as a function of $d_{raw}$ suggests that it serves as a *fuzzy mapping* of the distance in the raw embedding space.

Finally, we show that Raptor preserves class separability in the original DINOv2-L space. For a binary classification dataset, assume there are classes $\mathcal{A}, \mathcal{B}$, whose cluster centers are given by $\mu_A, \mu_B$. The cluster centers (in the DINOv2-L space) are set apart by $\|\boldsymbol{\mu}_A - \boldsymbol{\mu}_B\| \geq \beta$. Then the distance between the cluster centers are given by

$$\|\Phi(\boldsymbol{\mu}_A) - \Phi(\boldsymbol{\mu}_B)\| \geq \frac{(1-\varepsilon)}{D}\alpha_{min}\beta = c\beta$$

Where $c > 0$ is some constant. That is, the cluster centers remain $\Omega(\beta)$-separated even after aggregation and RP, thereby guaranteeing class separability. Empirically, we find that this is generally the case for the datasets we have benchmarked on, as Raptor generally exhibits superior performance in classification tasks.

However, when there is a violation in our assumptions (e.g., there is no slice-level differences in the embedding space between two volumes or such differences are not "smooth"), the lower-bound of the Raptor distance is no longer true (the upper bound still holds), and it may vanish to zero. We conjecture that this may be the case for datasets where Raptor performs relatively poorly (e.g., 3D FractureMNIST); in Figure A.3, we estimate the $\alpha_j$ values of volume pairs for the 3D MedMNIST datasets to bolster our point. As seen in the figure, there are a lot of $\alpha_j < 0$ in the axial view of the Fracture dataset. Intuitively, what this means is that some portion of $d_{Raptor}$ will be canceled out as we average over the slices, thereby reducing the distance between two semantically distinct volumes. For other datasets where this is not as significant, we conjecture that Raptor well approximates the distances in the raw embedding space, and results in a superior downstream performance.

*Figure A.3.* **Histogram of $\alpha_j$ estimates for pairs of volumes in each 3D MedMNIST dataset.** The $\alpha_j$ estimates for the fracture dataset slightly violates our core assumption of alignment ($\alpha_j > 0$), especially in the Axial view. This means that the Raptor distance $d_{Raptor}$ is no longer a good proxy for $d_{raw}$, as some slice-level embedding differences between volumes are canceled out in $d_{Raptor}$. The lowest $5\%$ quantile values of $\alpha_i$ is annotated as well. In many datasets, there are peaks of $\alpha_j = 0$, mostly due to the empty slices in the volumes.

## A.7. Runtime complexity of Raptor

We compute Raptor embeddings by taking the average of patch-level tokens inferred by a pretrained ViT in three orthogonal directions, then applying random projections to further compress the semantic information preserved in the tokens. Given $z \in \mathbb{R}^{3 \times D \times d \times p^2}$, the intermediate tensor inferred by applying the ViT to every observed image, we first apply the average step. For a dataset with $N$ volumes, reducing the sampling direction of each tensor (the dimension of size $D$) requires iteration over the tensor, incurring a cost of $\mathcal{O}(p^2 D d N)$. The resulting intermediate tensor is of size $3 \times d \times p^2$. Then, the projection step with random matrix $R$ which projects down the token dimension (of size $d$) incurs a cost of $\mathcal{O}(p^2 d K N)$. In total, the cost is $\mathcal{O}(p^2 D d N + p^2 d K N) = \mathcal{O}(p^2 d N (D + K))$.

## A.8. Comparison to Alternate Compression Methods

***Why not just use PCA?*** While PCA can also reduce dimensionality, it requires (1) data-dependent fitting, (2) storing the eigenvector matrix for all new data, and (3) does not come with distance-preservation guarantees for all points in general. PCA preserves variance in directions of largest eigenvalues but might discard subtle signals that are crucial for certain tasks. In contrast, random projection has uniform guarantees over all distances and does not require any optimization.

***Why not train a smaller parametric layer?*** Another option is to learn a small linear or MLP-based projection on top of the 256-d tokens. This typically requires a labeled set or at least a self-supervised objective, incurring additional training time and potential overfitting (especially in resource-scarce domains). By contrast, random projection is *train-free* and general-purpose.

***Why not use a nonlinear aggregation?*** A simple linear aggregation method allows for many "nice" properties, such as Lipschitz continuity and commutativity; not to reiterate the similar argument against training a nonlinear aggregator.

***Why is it necessary to aggregate over the slices?*** Because we use random projection to embed the local geometry of the tokens, it doesn't matter whether we aggregate over the slices or patches (or any similar ways), since with high probability, the pairwise distances of the volumes will be preserved (under certain assumptions, e.g., Appendix A.6).

### A.9. Dataset Characteristics

*Table A.5.* Summary of benchmarked datasets. For classification tasks, the balances of the classes are also reported.

| DATASET | SIZE | TASK | CLASSES | 1 | 2 | 3 | 4 | 5 | 6 | 7 | 8 | 9 | 10 | 11 |
|---|---|---|---|---|---|---|---|---|---|---|---|---|---|---|
| ORGAN | 1,742 | CLS | 11 | 11% | 11% | 11% | 11% | 11% | 10% | 10% | 10% | 4% | 3% | 3% |
| NODULE | 1,663 | CLS | 2 | 79% | 20% | | | | | | | | | |
| FRACTURE | 1,584 | CLS | 3 | 43% | 37% | 19% | | | | | | | | |
| ADRENAL | 1,370 | CLS | 2 | 76% | 23% | | | | | | | | | |
| VESSEL | 1,908 | CLS | 2 | 88% | 11% | | | | | | | | | |
| SYNAPSE | 1,759 | CLS | 2 | 73% | 26% | | | | | | | | | |
| CC-CCII | 2,471 | CLS | 3 | 41% | 36% | 21% | | | | | | | | |
| CTRG-C | 1,804 | CLS (MULTI) | 4 | 98% | 87% | 67% | 29% | | | | | | | |
| CTRG-B | 6,000 | CLS (MULTI) | 3 | 92% | 61% | 13% | | | | | | | | |
| WHITE MATTER | 1,476 | REG | 4 | | | | | | | | | | | |
| GRAY MATTER | 1,476 | REG | 6 | | | | | | | | | | | |
| CEREBELLUM | 1,476 | REG | 28 | | | | | | | | | | | |
| AMYGDALA | 1,476 | REG | 8 | | | | | | | | | | | |
| HIPPOCAMPUS | 1,476 | REG | 8 | | | | | | | | | | | |
| CORTEX | 1,476 | REG | 34 | | | | | | | | | | | |
| GYRUS | 1,476 | REG | 50 | | | | | | | | | | | |
| PALLIDUM | 1,476 | REG | 8 | | | | | | | | | | | |
| CAUDATE | 1,476 | REG | 8 | | | | | | | | | | | |
| THALAMUS | 1,476 | REG | 8 | | | | | | | | | | | |

## A.10. Additional Results

*Table A.6.* AUPR (precision-recall) for all classification benchmarks.

| METHODS | ORGAN | NODULE | FRACTURE | ADRENAL | VESSEL | SYNAPSE | CC-CCII | CTRG-C | CTRG-B |
|---------|-------|--------|----------|---------|--------|---------|---------|--------|--------|
| MISFM   | 0.940 | 0.850  | 0.546    | 0.853   | 0.880  | 0.873   | 0.934   | 0.658  | 0.659  |
| SUPREM  | 0.994 | 0.841  | 0.484    | 0.883   | 0.910  | 0.892   | 0.978   | 0.704  | 0.711  |
| VOCO    | 0.943 | 0.780  | 0.544    | 0.884   | 0.656  | 0.831   | 0.811   | 0.634  | 0.648  |
| MERLIN  | 0.845 | 0.784  | 0.522    | 0.803   | 0.718  | 0.821   | 0.874   | 0.679  | 0.647  |
| **RAPTOR** | 0.994 | 0.884 | 0.503 | 0.889   | 0.906  | 0.929   | 0.990   | 0.704  | 0.688  |

*Table A.7.* (Micro averaged) AUPR (precision-recall) for all classification benchmarks.

| METHODS | ORGAN | NODULE | FRACTURE | ADRENAL | VESSEL | SYNAPSE | CC-CCII | CTRG-C | CTRG-B |
|---------|-------|--------|----------|---------|--------|---------|---------|--------|--------|
| MISFM   | 0.922 | 0.937  | 0.509    | 0.761   | 0.970  | 0.536   | 0.913   | 0.776  | 0.819  |
| SUPREM  | 0.995 | 0.942  | 0.487    | 0.940   | 0.986  | 0.925   | 0.983   | 0.846  | 0.914  |
| VOCO    | 0.955 | 0.893  | 0.543    | 0.943   | 0.937  | 0.890   | 0.776   | 0.821  | 0.898  |
| MERLIN  | 0.865 | 0.888  | 0.575    | 0.895   | 0.960  | 0.887   | 0.876   | 0.824  | 0.893  |
| **RAPTOR** | 0.994 | 0.956 | 0.552 | 0.886   | 0.983  | 0.959   | 0.990   | 0.846  | 0.917  |

*Table A.8.* (Micro averaged) AUROC scores for all classification benchmarks.

| METHODS | ORGAN | NODULE | FRACTURE | ADRENAL | VESSEL | SYNAPSE | CC-CCII | CTRG-C | CTRG-B |
|---------|-------|--------|----------|---------|--------|---------|---------|--------|--------|
| MISFM   | 0.988 | 0.936  | 0.708    | 0.749   | 0.972  | 0.442   | 0.945   | 0.720  | 0.765  |
| SUPREM  | 0.999 | 0.941  | 0.682    | 0.937   | 0.986  | 0.929   | 0.990   | 0.802  | 0.902  |
| VOCO    | 0.994 | 0.903  | 0.716    | 0.940   | 0.944  | 0.893   | 0.867   | 0.780  | 0.886  |
| MERLIN  | 0.982 | 0.905  | 0.731    | 0.901   | 0.959  | 0.889   | 0.934   | 0.794  | 0.884  |
| **RAPTOR** | 0.999 | 0.954 | 0.709 | 0.900   | 0.982  | 0.960   | 0.995   | 0.803  | 0.904  |

