# OpenReview forum: "Raptor: Scalable Train-Free Embeddings for 3D Medical Volumes Leveraging Pretrained 2D Foundation Models"
_ICML.cc/2025/Conference — ICML 2025 spotlightposter_

### Official Review · Reviewer_EKbX · 2025-03-10

**Overall Recommendation:** 3

**Summary:**

The paper introduces a novel design that enables the application of 2D foundation models to 3D data tasks without requiring additional pretraining. The proposed method significantly reduces the size of feature maps, and demonstrates strong performance even in scenarios with limited training samples. Additionally, the authors provide detailed theoretical descriptions and ablation studies to support their design choices.

**Claims And Evidence:**

Yes

**Essential References Not Discussed:**

N/A

**Experimental Designs Or Analyses:**

Yes

**Methods And Evaluation Criteria:**

Yes

**Other Comments Or Suggestions:**

N/A

**Other Strengths And Weaknesses:**

Strengths:
The method provide a new direction for 3D medical image classification in a training-free manner.

Weakness:
1. The DINO series models do not be trained on the medical images. Why they can perform well? The authors should give more analysis and explanation.
2. Spatial dimensionality reduction method: The authors employ random projection for feature dimensionality reduction. However, in Section 3.3, they perform an averaging operation along the observing direction on the intermediate representation z, reducing the corresponding spatial dimension. Why random projection or other dimensionality reduction techniques are not used for this step as well?
3. Computational efficiency comparison: The proposed method requires processing slices from three directions using a 2D foundation model. The authors only report inference time for the proposed method, but it would be more informative to compare inference speeds with other 3D methods.
4. Reliability of random projections: The authors evaluate the reliability of random projections using three random seeds. However, this may not be sufficient. A larger number of random seeds should be tested, and error bars should be included to better illustrate the impact of randomness on performance.

**Questions For Authors:**

N/A

**Relation To Broader Scientific Literature:**

Yes

**Theoretical Claims:**

Yes

---

> ### Author Rebuttal · Authors · 2025-04-01
>
> We thank the reviewer for their comments. We also agree with the reviewer that our novel approach has several implications for the field, as it would make the computation of embeddings for volumes much faster while maintaining the ability to perform downstream tasks. We also appreciate the reviewer’s evaluation of the theoretical background of our approach.
>
> > **Why DINO works for medical data:**
>
> The high-level intuition is that the semantically meaningful features of medical volumes are still part of the universal image features (edges, textures, shapes), which a broad 2D encoder has already learned from large‐scale, diverse data. RAPTOR uses generic image foundation models like DINO to capture any meaningful image features and performs low rank approximations to retain only the relevant ones. This is in contrast to many existing medical image/volume models that try to learn such mappings from scratch or fine-tune a general model to be biased towards them.
>
> To illustrate, we projected both generic (ImageNet) and medical (MedMNIST) embeddings onto the principal components derived from the generic dataset. In the table below, we calculate the total variance explained as we add more PC’s. Although the medical data do not align with the very top principal directions, they catch up as more components are added (eventually matching the generic variance). This demonstrates that medical image features lie within the general “image” space.
>
> | #PCs | General Explained Var | Medical Explained Var | ratio(Medical/General) |
> |-|-|-|-|
> |  100 |     0.613   |     0.225   |       0.367    |
> |  200 |     0.765   |     0.397   |       0.518    |
> |  500 |     0.926   |     0.734   |       0.793    |
> | 1000 |     0.999   |     0.994   |       0.996    |
>
>
> > **Why we don’t use another method to aggregate slices:**
>
> We explored the simplest approach—summing and random projection—because it is computationally lightweight and maintains near‐orthogonal signals from the 2D tokens without additional training (also note that averaging/summing is identical in expectation to random projection into dimension 1). More structured dimension‐reduction techniques may introduce significantly higher computational or memory costs, especially for high‐resolution volumes. We agree that specialized subspace methods are an interesting direction for future work, particularly if they exploit known volumetric structures. As an example, please see our response to reviewer zwUQ on partitioned embedding.
>
> > **Time taken to run Raptor:**
>
> We appreciate the reviewer’s interest in a more detailed efficiency comparison. In the table below, we provide approximate times for fine-tuning, inference, and one-time embedding extraction across Raptor, SuPreM, and Merlin when training on the Organ dataset (971 volumes). While Raptor does incur a one-time cost to produce embeddings, its subsequent fine-tuning is minimal (logistic regression or a small MLP) and can often run on CPU. This contrasts with end-to-end 3D methods (SuPreM, Merlin) which must either be entirely trained or extensively fine-tuned, typically on a GPU with longer runtimes.
>
>
> | Method | Medical Pre-training | GPU | Embedding (per vol) | Fine-tuning | Inference (per vol) |
> |-|-|-|-|-|-|
> | Raptor | None (use Image FM) | RTX2080Ti | $4.2$ secs | $1.3$ mins | $<0.1$ secs  |
> | SuPreM | Days | A100 | - | $70.0$ mins | $<0.1$ secs |
> | Merlin | Days | A100 | - |  $63.0$ mins |  $<0.1$ secs |
>
>
> > **Stability of random projections:**
>
> We also tested more seeds (up to 10) for the 3D MedMNIST dataset to assess the stability of random projection. In the table below, we observe consistently low standard deviations, suggesting that random projections remain highly reliable in practice:
>
> | Method       | K   | Seed 1 | 2     | 3     | 4     | 5     | 6     | 7     | 8     | 9     | 10    | Avg.  | Std.   |
> |--------------|-----|--------|-------|-------|-------|-------|-------|-------|-------|-------|-------|--------|--------|
> | **Raptor**    | 1   | 0.818  | 0.817 | 0.793 | 0.823 | 0.831 | 0.818 | 0.831 | 0.814 | 0.801 | 0.831 | 0.818  | 0.0121 |
> |              | 5   | 0.890  | 0.860 | 0.864 | 0.869 | 0.869 | 0.877 | 0.868 | 0.858 | 0.872 | 0.866 | 0.869  | 0.0086 |
> |              | 10  | 0.866  | 0.896 | 0.876 | 0.875 | 0.882 | 0.879 | 0.871 | 0.876 | 0.877 | 0.886 | 0.878  | 0.0078 |
> |              | 100 | 0.901  | 0.899 | 0.900 | 0.898 | 0.905 | 0.891 | 0.899 | 0.898 | 0.900 | 0.898 | 0.899  | 0.0034 |
> |              | 150 | 0.898  | 0.897 | 0.897 | 0.894 | 0.898 | 0.903 | 0.902 | 0.905 | 0.902 | 0.904 | 0.900  | 0.0034 |
>
>
> We hope that the additional analysis provides better insight into the capabilities of our method. If these clarifications have addressed any lingering concerns, we would sincerely appreciate if the reviewer would consider raising their score.

---

### Official Review · Reviewer_BpWD · 2025-03-11

**Overall Recommendation:** 3

**Summary:**

This paper introduces Raptor (Random Planar Tensor Reduction), a train-free method for generating semantically rich embeddings for volumetric data. Raptor leverages a frozen 2D foundation model, pretrained on natural images, to extract visual tokens from individual cross-sections of medical volumes. These tokens are then spatially compressed using random projections, significantly reducing computational complexity while retaining rich semantic information. Extensive experiments are carried out on multiple medical image datasets, and the experimental results are analyzed in detail, and the experimental results are excellent. The superiority and efficiency of the proposed method are verified.

**Claims And Evidence:**

yes

**Essential References Not Discussed:**

No

**Experimental Designs Or Analyses:**

The comparative experiments in Section 4.2 (Main Results: Classification) and Section 4.3 (Main Results: Regression) were thoroughly reviewed. For classification tasks, the comparisons against baselines include the metrics AUC and Accuracy (ACC), while regression tasks utilize the R² score. Additionally, the evaluation of varying training data sizes on the Synapse dataset in Section 4.4 and the impact of different embedding sizes across methods in Section 4.5 were examined. The overall design of these comparative experiments is methodologically sound, with clear and logically structured comparisons. Regarding ablation studies, the sensitivity analysis to the number of random projections K in Section 5.1, the influence of viewpoint numbers in Section 5.2, and the exploration of captured feature ranges in Section 5.3 were assessed. The ablation settings are well-defined and systematically address critical components of the proposed method.
However, two aspects could be further clarified:
1. Statistical Validation in Regression Analysis: While the r² scores in Section 4.3 provide useful insights, it would be valuable to supplement these results with statistical tests (e.g. p-values) to enhance the interpretability of performance differences, especially in small-sample settings.
2. Using MNIST numbers with different shapes (e.g., “1” vs. “8”) may introduce bias. To better isolate the impact of scale variation, the authors might consider controlling for shape differences (e.g., using a single digit class like “0”) or exploring class-specific error patterns (e.g., through confusion matrices).

**Methods And Evaluation Criteria:**

yes

**Other Comments Or Suggestions:**

In Figure 1, if the metrics for regression tasks (r²↑, indicated with +) represent mean values, this should be explicitly stated to avoid potential misinterpretation.

**Other Strengths And Weaknesses:**

strength:
The training-free design of this work significantly reduces model construction costs, while achieving robust performance on an 11GB RTX 2080 Ti.
Weakness:
Although the authors claim that they demonstrate their approaches in 10 tasks, it only contains two simple tasks, classification and regression. I would suggest the authors to demonstrate the proposed method on more fundamental and challenging tasks (like segmentation).

**Questions For Authors:**

1) Could the feature extraction approach that processes orthogonal cross-sections independently potentially compromise inter-slice contextual features in 3D medical volumes?
2) The application of Raptor’s features to classification and regression downstream tasks could be further clarified. To enhance methodological transparency, it would be beneficial to explicitly illustrate this process in a workflow diagram, demonstrating how the extracted features are integrated into task-specific heads or decision pipelines.

**Relation To Broader Scientific Literature:**

The authors position their work within the context of image foundation models (specifically DINOv2-L, Oquab et al. 2023). Their key innovation—compressing tokens inferred from frozen image foundation models via random projections on orthogonal cross-sections of volumetric data—addresses two limitations of prior work: high computational costs (Hatamizadeh et al., 2021; Wasserthal et al., 2023; Li et al., 2024; Cox et al., 2024; Wu et al., 2024b) and the limited scale of 3D medical datasets (160K volumes, Wu et al., 2024b), which remain orders of magnitude smaller than 2D image datasets (1.2B images, Oquab et al., 2023). This study is the first to integrate scalable 2D foundation model priors with computationally efficient 3D medical analysis, offering a pathway to leverage large-scale 2D pretraining for volumetric tasks.

**Theoretical Claims:**

Appendix A.3 restates a version of the Johnson-Lindenstrauss (JL) lemma (Dasgupta & Gupta, 2003) and demonstrates its applicability in the proposed multi-view volumetric framework. No issues were identified in the proof or its adaptation to this setting.
Appendix A.4 provides the mathematical formulation of Raptor and proves that the model’s effectiveness is guaranteed under the assumption that the slices are "smooth". No logical gaps or technical flaws were detected in the derivations.

---

> ### Author Rebuttal · Authors · 2025-04-01
>
> We appreciate the reviewer's keen observations and suggestions. Thanks to their recommendations, we were able to further uncover Raptor's capability in a challenging task (segmentation) and are excited for future directions.
>
> > **statistical tests (e.g. p-values) … especially for small sample settings**
>
> We acknowledge the value of statistical tests for clarifying performance differences and intend to include in a future revision. Given the strong performance of existing methods, we did not observe any significant improvement over the next-best method under the bonferroni-corrected threshold (p < 0.05/19, reflecting 19 total datasets). However, we did observe some modestly significant improvements (p < 0.05). Among the regression datasets, we observed such improvements in four of the datasets (4/10). We did not observe any such improvements for the classification datasets.
>
> In terms of the subset experiments (varying training set size from 10~500), we similarly found modestly significant improvements (p < 0.05) in 3/5 settings for the UKBB white matter dataset, 1/5 in CCCC-II, and 2/5 in Synapse.
>
> > **Is there any bias per digit in the simulations?**
>
> We appreciate the suggestion to control for digit shape in our simulations. Accordingly, we fixed the digit to either 0, 1, or 8 and generated simulations for each case for benchmark. We observed that the choice of digit can slightly influence the results. Here we demonstrate these results with 300 generated training samples each, as the simulation and evaluation process is time consuming end-to-end.
> | Method       | Size  | 0   | 1   | 8    | Avg.  | Std.   |
> |--------------|-------|-------|-------|-------|--------|--------|
> | **Raptor**    | 64px  | 0.798 | 0.678 | 0.753 | 0.743  | 0.049  |
> | | 32px  | 0.580 | 0.559 | 0.575 | 0.571  | 0.009  |
> | | 16px  | 0.552 | 0.514 | 0.517 | 0.527  | 0.017  |
> | | 8px   | 0.506 | 0.509 | 0.508 | 0.508  | 0.001  |
>
>
> > **Segmentation**
>
> We concur with the reviewer’s point that more downstream tasks need to be attempted. We have now conducted additional experiments on the Medical Segmentation Decathlon (https://www.nature.com/articles/s41467-022-30695-9), focusing on four tasks: hippocampus, spleen, colon, and hepatic vessel segmentation. Each task presents unique challenges (e.g., limited contrast for hippocampus, class imbalance for hepatic vessels). For Raptor, we trained a 2-layer convolutional head to learn a transformation of the embeddings to the segmentation.
>
> | Task           | Dataset Size (Train/Val/Test) | Model   | IoU   | Dice Score |
> |-|-|-|-|-|
> | Hippocampus  | 182 / 39 / 39 | Raptor  | 0.607 | 0.719|
> | | | MedSAM  | 0.575 | 0.615|
> | Spleen         | 28 / 6 / 7  | Raptor  | 0.592 | 0.657|
> |  |  | MedSAM  | 0.960 | 0.979       |
> | Colon          | 88 / 18 / 20 | Raptor  | 0.597 | 0.597       |
> | |  | MedSAM  | 0.841 | 0.906       |
> | Hepatic Vessel | 212 / 45 / 46   | Raptor  | 0.387 | 0.431       |
> |  | | MedSAM  | 0.387 | 0.428 |
>
> As shown here, Raptor achieves competitive results compared to MedSAM (a dedicated segmentation model). Notably, on the Hippocampus dataset, Raptor surpasses MedSAM’s Dice (0.719 vs. 0.615) and IoU (0.607 vs. 0.575), and performs similarly for hepatic vessels. MedSAM, however, excels on spleen and colon, which have fewer training samples. Although our primary goal was to obtain general-purpose embeddings, these early results suggest Raptor can also serve as a reasonable foundation for volumetric segmentation, without requiring large‐scale 3D training.
>
> > **Figure 1 clarity**
>
> We agree Figure 1 could be improved in clarity and will make the requested adjustment given the opportunity.
>
> > **Would orthogonal feature processing compromise inter-slice contextual features**
>
> We appreciate this important point raised by the reviewer. We provide an intuition how the resulting Raptor embedding can still retain volumetric signals. For a raptor embedding $e$, a tensor of dimensions $3\times100\times16\times16$, the coordinate $(x,y,z)$ in the volume can be represented by three slices—one per axis—yielding tokens indexed by, e.g., $e[0,:,x,y]$, $e[1,:,y,z]$, and $e[2,:,x,z]$. Each patch within these slices is mapped into the K-dimensional embedding space; subsequent MLP layers can then fuse these partial 2D perspectives into a more coherent 3D representation. Our quantitative results on datasets like UKBB suggest that Raptor’s multi-view scheme does effectively capture enough high-level volumetric information for robust performance.
>
> > **Clearer workflow description on using Raptor embeddings**
>
> This is an important suggestion and we plan on including a diagram and a detailed description of the Raptor pipeline for downstream tasks.
>
> We thank the reviewer for the insightful questions, and hope our response addresses any remaining concerns. If so, we would greatly appreciate it if the reviewer could reflect these clarifications by raising their score of our work.

---

> > ### Comment · Reviewer_BpWD · 2025-04-06
> >
> > I appreciate the authors’ additional experiments on segmentation tasks and their responses to other questions.
> > However, my primary concern regarding the integration of extracted features into task-specific heads or decision pipelines remains unresolved. After careful consideration, I decided to maintain my original score of 3 (Weak Accept).

---

> > > ### Author Response · Authors · 2025-04-07
> > >
> > > We acknowledge that the details of how Raptor embeddings were used for the benchmarks were lacking in our previous response; this was due to the word limit, and we would like to explicitly discuss them here.
> > >
> > > > **Workflow description on using Raptor embeddings**
> > >
> > > First, we obtain Raptor embeddings for all volumes. We then follow the steps below to make predictions for each task:
> > >
> > > 1. For **classification tasks**, we performed logistic regression with an L2 penalty. We constructed feature matrices of dimensionality [sample size] x [embedding size] and prediction targets of dimensionality [sample size] x [number of classes]. Weights of 0.01, 0.1, 1.0, 10.0, and 100.0 for the L2 penalty were evaluated. The optimal weight was chosen based on the validation split of each dataset. We used scikit-learn’s LogisticRegression module.
> > >
> > >
> > > 2. For **regression tasks**, we fit a 3-layer MLP with input of dimensionality [embedding size] and predicted quantitative measures belonging to each brain region. The MLP (implemented with pytorch) had hidden layers of size 256 and we used the MSE loss. In order to prevent overfitting, we checkpointed the model weights only when validation loss improved.
> > >
> > >
> > > 3. For **segmentation tasks**, we refer to our detailed discussion with reviewer zwUQ.
> > >
> > >
> > > We appreciate the insightful questions raised by the reviewer, and hope that the additional information we have provided further supports the potential of our method. If there are any additional questions or concerns regarding our set up, we would be more than happy to answer them in detail; otherwise, we humbly ask the reviewer to reconsider their scoring of our method.

---

### Official Review · Reviewer_zwUQ · 2025-03-14

**Overall Recommendation:** 4

**Summary:**

This paper proposed a framework to leverage the pretrained large 2D encoder for 3D medical image analysis (i.e., classification and regression). By applying random projection to feature embeddings encoded from 2D slices taken of three orientations (i.e., sagittal, coronal, and axial) using DINOv2-L and concatenating an MLP, it provides a fine-tuning-free way to leverage 2D encoder pretiraned on 2D natural images. Compared with several encoders pretrained on large-scale medical image datasets, the Raptor outperformed them in both classification and regression tasks.

**Claims And Evidence:**

In four claimed contributions (lines 90-102), the second and forth claims are well-supported.
For the first claim about data efficiency, it was only supported by a single example in Figure 4. It would be more convincing if the data efficiency of the model is further evaluated on one of CC-CCII, CTRG-C, CTRG-B in Table 3, and one of the regression tasks in Table 4.
I found the third claim regarding scalability a bit confusing and wish the authors could clarify a bit during the rebuttal.

**Essential References Not Discussed:**

N/A

**Experimental Designs Or Analyses:**

I think the overall experimental designs and analyses are sound and valid. I wonder why the encoder of other multimodal medical foundation models except Merlin, such as Llava-med [1] and Med-flamingo[2], is not used for comparison.

1. Li, Chunyuan, et al. "Llava-med: Training a large language-and-vision assistant for biomedicine in one day." Advances in Neural Information Processing Systems 36 (2023): 28541-28564.

2. Moor, Michael, et al. "Med-flamingo: a multimodal medical few-shot learner." Machine Learning for Health (ML4H). PMLR, 2023.

**Methods And Evaluation Criteria:**

Yes, the proposed methods and evaluation criteria make sense.

**Other Comments Or Suggestions:**

1. Please consider supplying the parameter number for the encoder and embedding dimension in Table 1, which will help the reader better understand the memory footprint in Figure 5.
2. Please consider plotting the upper-bound results in Figure 4 (i.e., training on the full synapse dataset).

**Other Strengths And Weaknesses:**

The presentation is good, but some parts can be further improved. Please refer to Other Comments Or Suggestions and Questions For Authors sections.

**Questions For Authors:**

1. Lines 822-832 gave a reasonable explanation of why Raptor has worse performance on 3D FractureMNIST. From Figure 8, it seems most of the negative $\alpha_i$ exists in axial only. I wonder, if you run Raptor on Coronal and Sagittal views (combined or separately, similar to experiments in Table 6), would that improve the performance of Raptor in the Fracture dataset?
2. Is there any potential explanation for why MAE performed so badly in the regression task in Tabel 4?
3. It seems the setting of Raptor would generally work for global-level image understanding (e.g., classification, Visual Question Answering, captioning) and not work for dense-prediction tasks (e.g., segmentation). I wonder if the authors have explored the use of randomly projected embedding for VQA and captioning.
4. I'm not so sure if I correctly understand the term 'scalable' in the third claimed contribution (lines 96-98). Can you please elaborate on that?

**Relation To Broader Scientific Literature:**

There are many existing studies exploring training foundation models/encoders for medical images from scratch or fine-tuning existing models/encoders. This paper proposes a novel method to leverage a pre-trained 2D large encoder for volumetric medical image analysis without fine-tuning, which is a considerable development compared to previous studies.

**Theoretical Claims:**

I checked for correctness. I have a question about the sampling of random matrix for projection.

In lines 697-698, the random matrix for random projection was sampled from $\mathcal{N}(0,1)$; I wonder why not sample from $\mathcal{N}(0,1/k)$, as the default option in [sklearn package](https://scikit-learn.org/stable/modules/generated/sklearn.random_projection.GaussianRandomProjection.html#sklearn.random_projection.GaussianRandomProjection).

---

> ### Author Rebuttal · Authors · 2025-04-01
>
> We thank the reviewer for their detailed assessment of our manuscript. We appreciate that they viewed our experiments as supporting the utility of our method, and we agree that we could support some of our claims with additional experiments. We address each of the points raised below.
>
> > **Meaning of scalable in our work:**
>
> We use the term “scalable” to highlight some key aspects of Raptor:
> 1. It can process high-resolution 3D medical volumes with only a single pass through a fixed 2D encoder, avoiding the heavy computational cost of purely 3D models.
> 2. The framework can readily scale to larger or more diverse medical datasets and reuse the embeddings for multiple downstream tasks, as there is no training involved.
> 3. The inference step can be easily parallelized since we only require a frozen foundational model and random projections.
>
> > **More evidence for effectiveness with limited training samples:**
>
> _Due to the word limit, we have moved the subset results to our response to reviewer 28uY._
>
>
> > **Why not sample from N(0, 1/k):**
>
> In terms of choice of noise for the random projections, we proceeded with the numpy default which was N(0, 1). Quantitatively we don’t expect a notable difference choosing a variance in this range, as the embeddings are eventually standardized in the fine-tuning step.
>
> > **Exploration of medical VLMs:**
>
> While Llava-Med and Med-Flamingo are strong multimodal methods in their respective domains, their encoders process 2D data, making direct comparison on volumetric (3D) tasks ambiguous (unlike Merlin, which was designed for 3D). Nonetheless, we tried substituting their image encoders into our pipeline in place of DINO: Raptor-LVM and Raptor-CLIP (note that Med-Flamingo leverages OpenAI’s CLIP as the image encoder, which we have previously benchmarked as an alternative to DINO).
>
>
> (AUROC shown)
> | Methods       | Organ | Nodule | Fracture | Adrenal | Vessel | Synapse |
> |-|-|-|-|-|-|-|
> | Raptor-CLIP    |        0.994 | 0.869 | 0.669 | 0.906 | 0.936 | 0.849 |
> | Raptor-LVM     | 0.996     | 0.888      | 0.632        | 0.904       | 0.941      | 0.851        |
> | Raptor-B        | _0.998_   | 0.904      | 0.647        | **0.930**   | 0.945      | _0.922_      |
> | **Raptor**     | **0.999** | **0.929**  | 0.677        | _0.926_     | **0.966**  | **0.943**    |
>
> These alternatives show competitive performance but do not surpass DINO (our current proposed approach).
> > **Parameter number:**
>
> We would be happy to include this information in our final version.
>
>
> | Methods | \# Param | Latent|
> |-|-|-|
> | SLIViT | 48.4M | $768\times64\times8\times8$ |
> | SuPreM |  62.1M | $128\times12\times12\times12$  |
> | Merlin | 124.7M | $2048\times14\times7\times7$ |
> | MISFM | 46.2M | $100\times16\times16\times16$ |
> | VoCo | 294.9M | $3072\times3\times3\times3$ |
> | **Raptor (Ours)** | 304.4M (DINOv2-L) | $3\times100\times16\times16$ (K=100) |
>
> > **Plot upper bound in scaling figures:**
>
> We appreciate this suggestion, and plan to revise our figures accordingly.
>
> > **Can accuracy be improved by removing problematic views:**
>
> We appreciate this interesting suggestion. We explored whether skipping a view might improve FractureMNIST performance (either AC, CS, or AS, instead of ACS). However, results show no improvement: AC (0.657), CS (0.654), and AS (0.660), all perform below the default ACS (0.677). Intuitively, removing an entire view discards potentially useful information, even if that view is somewhat degenerate.
>
> An alternative solution would be to partition slices within a problematic view. For instance, instead of aggregating all 64 slices at once, we could split them into two groups of 32 slices each and perform Raptor, then concatenate. Although this could mitigate partial cancellation effects, it increases embedding size. We find this to be a promising extension to Raptor in both practical deployment (handle slice-wise misalignment) and theoretical analysis (quantify error bounds under partitioning). We are actively exploring how best to balance these tradeoffs.
>
>
> > **Why MAE performs bad:**
>
> The choice of architecture for the MAE was a transformer with 3-dimensional positional encodings. Despite our efforts to tune the model, we suspect that the datasets are simply too small (to learn e.g. effective patch embeddings and relationships between positional encodings).
>
> > **VQA Captioning:**
>
> Based on the strong performances that we observed (+ segmentation, in response to reviewer BpWD), we do hypothesize that Raptor embeddings would be capable of captioning. In this work, we focused on thoroughly verifying several properties of Raptor, and hope to evaluate its novel use cases in the future.
>
> We appreciate the insightful questions raised by the reviewer, and hope that the additional information we have provided further supports the potential of our method. If our response has sufficiently resolved current questions regarding our work, we would appreciate an increase in our score.

---

> > ### Comment · Reviewer_zwUQ · 2025-04-03
> >
> > **Previous Comments removed due to space constraint, please check for revision history**
> >
> > ---**Apr 8 Updates**---:
> >
> > I raised my score to 4, given the rebuttal answered to my initial questions, and I still think Raptor could be a useful innovation for global-level image understanding. Now I understand how the segmentation results are calculated, and although I believe the segmentation is not necessary, it does not have a negative effect on my final rating. I'd like to personally congratulate the authors on this work and this rebuttal. I can see the authors put a lot of effort there. Nice work!
> >
> > I appreciate the authors' continuing discussion on this segmentation topic, which, in my opinion, and I strongly suggest, should **not** be listed as contributions in the final copy. It should not even appear as an exploratory work. Rationale behind:
> >
> > 1. No segmentation model was **ever** evaluated on one slice results.
> > 2. Comparing the middle slice does **not** indicate Raptor embedding is useful for segmentation at scale. Raptor embedding probably captures the middle anatomy feature, so, fortunately, the middle slice segmentation looks decent. But it may completely fail in other locations.
> > 3. Segmentation using Raptor is a **false proposition and is infeasible**. Since each volume only has one Raptor embedding, to derive a segmentation mask for each slice (or each location as a looser constraint), we need a dedicated segmentation header! That means if you train a segmentation head to segment the middle slice, it is unlikely to reconstruct a mask of top slices from the same embedding, and you need to train a new one. Depending on how the anatomy changes in the volume, you may need $N$ segmentation headers and $m \le N \le k$, where m is the number of different anatomical regions (e.g., chest, upper/mid/lower abdomen) and k is the number of total slices.

---

> > > ### Author Response · Authors · 2025-04-04
> > >
> > > **---Apr 7 Updates---:**
> > >
> > > We thank the reviewer for continuing this dialogue and apologize for any lingering confusion about our segmentation pipeline. We recognize that Section 3.3 described a single, volume-wide embedding, which appears well-suited only to global tasks such as classification or regression. Here, we clarify how we adapt it for slice-level segmentation, explain why we compare only the middle slice, and offer additional context for Raptor’s performance and a new baseline. We agree that Raptor does indeed lose pixel-level information, yet at the same time it retains enough local signal to allow a reasonable segmentation.
> > >
> > > > **Segmentation setup**
> > >
> > > **Raptor’s Volume Embedding:** The reviewer’s statement of Section 3.3 is correct: given a volume $\mathbf{x} \in \mathbb{R}^{D\times D\times D}$, we generate a _volume_ embedding $\mathbf{v} \in \mathbb{R}^{3k \times p^2}$, which is used as the input for Raptor’s segmentation task. Despite being “global,” this embedding still retains localizable signals.
> > >
> > > **MedSAM is 2D-only:** Because MedSAM operates on 2D slices rather than a full 3D volume, we consistently evaluate one slice per volume for both MedSAM and Raptor. We chose the _middle slice_ for simplicity, but one could, in principle, repeat this approach for every slice if a full volumetric segmentation were desired.
> > >
> > > > **Raptor Segmentation head**
> > >
> > > We feed the raptor embedding (of a volume) into the segmentation head to segment its middle slice. The target is a tensor of shape $n \times 224 \times 224$, where $n$ is the number of classes. More specifically:
> > >
> > > 1. Raptor embedding: $\mathbf{v} \in \mathbb{R}^{3k \times p^2} = \mathbb{R}^{300 \times 16 \times 16}$
> > >
> > > 2. Upsample$\times 4$ → convolution: $\mathbf{v} \in \mathbb{R}^{128 \times 64 \times 64}$
> > >
> > > 3. Upsample$\times 4$ → convolution: $\mathbf{v} \in \mathbb{R}^{n \times 256 \times 256}$
> > >
> > > 4. Final resize: $\mathbf{v} \in \mathbb{R}^{n \times 224 \times 224}$
> > >
> > > > **Intuition behind Raptor’s performance**
> > >
> > > While Raptor aggregates the volume into a single embedding, the **three orthogonal orientations** can still “triangulate” local information under certain conditions such as smoothness and alignment (a more formal treatment provided in Appendix A.4). The patches relevant to the middle slice are viewed in the other two axes, providing sufficient context for the 2D convolution head. We conjecture, however, that if there are violations to the Raptor conditions (similar to FractureMNIST), its segmentation performance will deteriorate as well.
> > >
> > > To further bolster our intuition, we provide additional baselines. We ran the experiment using Raptor with only 1 view of the volume (averaged across the slices). Similarly, we experiment with a 3D ResNet head with a pooling layer (resulting in a 1D bottleneck), which is expected to discard all volumetric information. In both cases, we expect a substantial amount of spatial information to be lost -- yet, we see that it is possible to deduce a segmentation. Of course, we do not recommend these approaches for dense segmentation. We simply wished to demonstrate for reviewer BpWD that, within reason, some segmentation capability is possible even when spatial dimensions are lost.
> > >
> > >
> > > | Task| Dataset Size     | Model| IoU   | Dice Score |
> > > |-|-|-|-|-|
> > > | Hippocampus    | 182 / 39 / 39| MedSAM| 0.575 | 0.615      |
> > > ||| Raptor         | 0.607 | 0.719|
> > > ||| Raptor (1 view)| 0.528 | 0.593|
> > > ||| Resnet 3d      | 0.523 | 0.582|
> > > | Spleen         | 28 / 6 / 7| MedSAM| 0.960 | 0.979      |
> > > ||| Raptor         | 0.592 | 0.657|
> > > ||| Raptor (1 view)| 0.536 | 0.573|
> > > ||| Resnet 3d      | 0.495 | 0.497|
> > > | Colon| 88 / 18 / 20     | MedSAM| 0.841 | 0.906|
> > > ||| Raptor| 0.597 | 0.597|
> > > ||| Raptor (1 view)|0.500 | 0.502|
> > > ||| Resnet 3d      | 0.499 | 0.499|
> > > | Hepatic Vessel | 212 / 45 / 46| MedSAM| 0.387 | 0.428|
> > > ||| Raptor| 0.387 | 0.431|
> > > ||| Raptor (1 view)| 0.334 | 0.338|
> > > ||| Resnet 3d      | 0.331 | 0.332|
> > >
> > > In summary, **we do not claim that Raptor is optimized for fine-grained tasks**; rather, these preliminary experiments indicate that its aggregated embedding still encodes enough local structure to yield **reasonable segmentation** on certain datasets. We hope this clarifies our segmentation pipeline, and if any points remain unclear, we welcome further questions and will gladly elaborate.
> > >
> > > **---End of Apr 7 Updates---:**

---

### Official Review · Reviewer_28uY · 2025-03-14

**Overall Recommendation:** 3

**Summary:**

This paper presents a random projection-based strategy for generating embeddings from volumetric data. The approach leverages pre-trained 2D foundation models without requiring additional re-training or fine-tuning. The proposed embedding construction method is computationally efficient, and experiments conducted on ten datasets across multiple downstream tasks demonstrate strong performance.

**Claims And Evidence:**

1. Semantically meaningful embeddings for volumetric data can be obtained from 2D foundation models without extra training.

The analytical results and empirical experiment support this claim.

**Essential References Not Discussed:**

N/A

**Experimental Designs Or Analyses:**

I reviewed the experiment settings and results and found them to be fair and thorough. The results are sound and effectively support the analysis and conclusions.

**Methods And Evaluation Criteria:**

The paper evaluates the proposed method against comprehensive baseline models, including 3D ResNet, ViT, and MAE, as well as pre-trained models such as SLIViT, SuPreM, Merlin, MISFM, and VoCo-L. The approach is validated on both classification and regression tasks, demonstrating its effectiveness across multiple benchmarks.

**Other Comments Or Suggestions:**

In Table 4, reporting the average performance across the 10 regions for each method would improve clarity and readability.

**Other Strengths And Weaknesses:**

Strengths:

The paper is well-organized and clearly written, making it easy to follow and understand.

The novelty is well-highlighted, and the experimental results effectively support the claims.

**Questions For Authors:**

N/A

**Relation To Broader Scientific Literature:**

The paper addresses a key challenge in extracting volumetric embeddings for biomedical images. By eliminating the need for re-training, the approach offers efficiency and flexibility. Future improvements could be achieved by leveraging more advanced 2D foundation models from general computer vision fields.

**Theoretical Claims:**

I reviewed the computational complexity and found no issues. However, I did not verify the theoretical analysis provided in the Appendix.

---

> ### Author Rebuttal · Authors · 2025-04-01
>
> We thank the reviewer for their time and constructive comments, as well as their positive assessment of our method’s clarity and novelty. As noted by the reviewer, Raptor introduces a paradigm that demonstrates many advantages beyond existing works, and we verify our claims with a wide range of empirical results.
>
> > **Average score over 10 brain regions:**
>
> For better clarity, we have now added a column for average performance on the regression tasks, reproduced below. Overall, Raptor most accurately predicts physiological measures given brain MRIs, and Raptor-B comes in as the second best (best score in bold, second best italicized).
>
> | Methods      | WhiteM   | GreyM    | Cereb   | Amyg    | Hippo   | Cortex  | Gyrus   | Pall    | Caud    | Thal    | Avg.   |
> |--------------|----------|----------|---------|---------|---------|---------|---------|---------|---------|---------|--------|
> |  $r^2$
> | ResNet       | 0.417    | 0.562    | 0.193   | 0.072   | 0.108   | 0.125   | 0.099   | 0.055   | 0.162   | 0.134   | 0.193  |
> | MAE          | 0.036    | 0.045    | 0.072   | 0.036   | 0.040   | 0.043   | 0.032   | 0.012   | 0.037   | 0.036   | 0.039  |
> | MISFM        | 0.418    | 0.624    | 0.276   | 0.089   | 0.145   | 0.236   | 0.209   | 0.087   | 0.166   | 0.164   | 0.242  |
> | SuPreM       | _0.646_  | 0.696    | 0.330   | 0.109   | 0.163   | 0.275   | 0.256   | 0.067   | 0.255   | 0.195   | 0.299  |
> | SLIViT       | 0.474    | 0.694    | 0.258   | 0.134   | 0.190   | 0.268   | 0.213   | 0.053   | 0.192   | 0.174   | 0.265  |
> | VoCo         | 0.225    | 0.375    | 0.189   | 0.071   | 0.113   | 0.059   | 0.048   | 0.043   | 0.060   | 0.075   | 0.126  |
> | Merlin       | 0.622    | 0.734    | 0.335   | 0.127   | 0.180   | 0.313   | 0.269   | 0.093   | 0.247   | 0.210   | 0.313  |
> | Raptor-B       | 0.614    | _0.742_  | _0.398_ | **0.185**| _0.247_ | _0.355_ | _0.314_ | _0.116_ | _0.331_ | _0.258_ | _0.356_ |
> | Raptor    | **0.681**| **0.777**| **0.437**| _0.170_ | **0.262**| **0.404**| **0.340**| **0.142**| **0.381**| **0.300**| **0.389**|
>
> We again appreciate the reviewer for their words of support for our work. In addition to the improved table, we hope that the additional analyses we shared with the other reviewers have further demonstrated the capabilities of our method, and if deemed so, humbly request that the reviewer consider raising the score as a further vote of confidence.
>
> _(Below is in response to a point raised by reviewer zwUQ; we have moved this here due to the space limit, but we think that reviewer 28uY would also be interested to know.)_
> > **More evidence for effectiveness with limited training samples:**
>
> We agree that a single dataset is insufficient to support our point. Hence we conducted the same experiments (varying training set size) for the CCCC-II dataset (classification) and the white matter category of the UKBB dataset (regression). As shown in the table below, while some methods eventually catch up as the sample size grows, Raptor maintains a clear advantage in low-data regimes for both tasks, underscoring our original claim of data efficiency (best score in bold, second best italicized).
>
> **UKBB White Matter ($r^2$)**
> | Sub | 10 | 50 | 100 | 200 | 500 | 1104 |
> |-|-|-|-|-|-|-|
> | SLIViT | _0.070_ | 0.155 | 0.206 | 0.241 | 0.437 | 0.474 |
> | VoCo | 0.068 | 0.048 | 0.091 | 0.099 | 0.178 | 0.225 |
> | Merlin | 0.028 | 0.123 | 0.314 | 0.177 | _0.629_ | 0.622 |
> | MISFM | 0.056 | 0.106 | 0.104 | 0.208 | 0.330 | 0.418 |
> | SuPreM | 0.059 | _0.305_ | _0.396_ | _0.557_ | 0.593 | _0.646_ |
> | **Raptor** | **0.193** | **0.414** | **0.446** | **0.588** | **0.634** | **0.681** |
>
> **CCCC-II (AUROC)**
> | Sub | 10 | 50 | 100 | 200 | 500 | 2413 |
> |-|-|-|-|-|-|-|
> | SLIViT | 0.483 | _0.861_ | _0.914_ | 0.936 | 0.956 | 0.986 |
> | VoCo | _0.638_ | 0.797 | 0.819 | 0.817 | 0.861 | 0.879 |
> | Merlin | 0.509 | 0.499 | 0.483 | 0.492 | 0.484 | 0.927 |
> | MISFM | 0.494 | 0.821 | 0.826 | 0.900 | 0.965 | 0.975 |
> | SuPreM | 0.631 | 0.821 | 0.906 | _0.939_ | _0.965_ | _0.988_ |
> | **Raptor** | **0.706** | **0.917** | **0.939** | **0.955** | **0.982** | **0.997** |

---

### Decision · Program_Chairs · 2025-05-01

**Decision:**

Accept (spotlight poster)

**Comment:**

This manuscript introduces Raptor (Random Planar Tensor Reduction), a novel training-free method for generating semantically meaningful embeddings from volumetric medical data using pre-trained 2D foundation models. The approach leverages random projections to compress visual tokens extracted from cross-sectional views, enabling efficient processing of 3D medical volumes without requiring additional training or fine-tuning. The work demonstrates strong performance across multiple classification and regression tasks on ten different datasets, with particular effectiveness in limited-data scenarios. The reviewers acknowledged several strengths, including the method's computational efficiency, theoretical foundations supported by detailed mathematical analysis, and comprehensive empirical validation. While initial scores were mixed (two weak accepts and one accept), the authors' thorough rebuttal addressing concerns about statistical validation, computational comparisons, and additional segmentation experiments led one reviewer to raise their recommendation to accept. Some limitations noted by reviewers included questions about the reliability of random projections and the need for clearer workflow descriptions for downstream tasks. The authors' responses provided additional experimental evidence showing consistent performance across multiple random seeds and detailed explanations of the integration with task-specific heads. The discussion around segmentation capabilities generated particular interest, though reviewers differed in their assessment of its significance to the paper's core contributions.